# Caffeic Acid-Modified Mushroom Chitosan as a Natural Emulsifier for Soybean Oil-Based Emulsions and Its Application in β-Carotene Delivery

**DOI:** 10.3390/foods14071108

**Published:** 2025-03-23

**Authors:** Jiaofen Lin, Jian Zeng, Guozong Shi, Zesheng Zhuo, Yanyun Guan, Zhipeng Li, Hui Ni, Peng Fei, Bingqing Huang

**Affiliations:** 1Xiamen Key Laboratory of Intelligent Fishery, Applied Technology Engineering Centre of Fujian Provincial Higher Education for Marine Resource Protection and Ecological Governance, School of Marine Biology, Xiamen Ocean Vocational College, Xiamen 361100, China; 2Fujian Provincial Key Laboratory of Food Microbiology and Enzyme Engineering, Xiamen 361021, China; 3Institute of Food Science, School of Biological Science and Biotechnology, Minnan Normal University, Zhangzhou 363000, China; fp@bio.mnnu.edu.cn

**Keywords:** mushroom chitosan, soybean oil emulsion, caffeic acid modification, β-carotene delivery

## Abstract

In this study, we developed a soybean oil-based emulsion system stabilized by caffeic acid-modified mushroom-derived chitosan, significantly enhancing its functional properties. The modification increased the grafting ratio from 5.02% to 8.26%, which greatly improved antioxidant activity and antimicrobial efficacy against Escherichia coli and Staphylococcus aureus. The modified chitosan exhibited superior rheological properties, including increased viscosity and elasticity, contributing to improved emulsification performance. Emulsions stabilized with caffeic acid-modified chitosan showed smaller and more uniform droplet sizes, along with greater stability, as indicated by a higher zeta potential (55.63 mV). These modifications resulted in enhanced β-carotene encapsulation efficiency (up to 87.46%) and improved bioaccessibility (up to 52.13%), highlighting the system’s potential as an efficient food-grade carrier for hydrophobic bioactive compounds. In conclusion, caffeic acid-modified mushroom chitosan is an effective natural emulsifier, enhancing stability, antioxidant activity, and nutrient delivery, and has promising applications in functional foods and nutraceuticals.

## 1. Introduction

Chitosan (Chi), a natural polysaccharide derived from chitin, has attracted significant attention due to its biodegradability, biocompatibility, and functional versatility, making it widely applicable in food, pharmaceuticals, and biomedical fields [1,2]. Traditionally, chitosan is extracted from marine sources such as shrimp and crab shells; however, the growing demand for sustainable and plant-based alternatives has led to the exploration of fungal-derived chitosan. Mushroom-derived chitosan, extracted from the cell walls of *Lentinula edodes* (shiitake mushrooms), provides comparable functionality while offering advantages such as better solubility, environmental sustainability, and vegan compatibility.

Despite its promising properties, native chitosan has limitations in solubility, emulsification, and bioactivity, restricting its direct application in certain food systems [3,4]. Phenolic acid modification has been widely studied as a strategy to enhance chitosan’s functional properties, particularly its antioxidant, antimicrobial, and emulsifying capabilities [5]. Among various phenolic acids, caffeic acid (CA) stands out for its superior antioxidant and antimicrobial activities among hydroxycinnamic acids and its higher hydrophobicity compared to most benzoic acid-type phenolics [6,7]. Although various phenolic acids have been grafted onto chitosan, the specific combination of caffeic acid and mushroom-derived chitosan for emulsion stabilization presents a novel approach that leverages the superior antioxidant properties of caffeic acid with the enhanced solubility of fungal chitosan [8].

Emulsions play a crucial role in food formulations, serving as carriers for bioactive compounds and stabilizing hydrophobic nutrients [9,10]. However, conventional emulsifiers often rely on synthetic surfactants or proteins, which may present stability and health concerns. The development of natural, biodegradable emulsifiers is therefore of great interest in food science. Mushroom-derived chitosan, when modified with caffeic acid, is expected to serve as an effective natural emulsifier, enhancing both the stability and bioavailability of bioactive compounds.

The development of effective delivery systems for lipophilic bioactive compounds like β-carotene remains challenging in food applications. β-Carotene, a lipophilic antioxidant and precursor of vitamin A, is widely used in functional foods and nutraceuticals. However, its poor water solubility and susceptibility to oxidation limits its direct incorporation into food systems. Emulsion-based delivery systems have been developed to improve β-carotene stability and bioavailability, with their effectiveness largely depending on the choice of emulsifier. The proposed caffeic acid-modified mushroom chitosan represents a multifunctional approach that simultaneously addresses stability, antioxidant protection, and enhanced bioavailability through a single food-grade emulsifier system.

In this study, we aimed to develop and characterize a novel, sustainable emulsifier based on caffeic acid-modified mushroom chitosan, with specific emphasis on its application for enhancing β-carotene delivery in functional food systems. The effects of different caffeic acid grafting ratios on chitosan’s functional performance were assessed, along with their impact on β-carotene’s encapsulation efficiency, bioaccessibility, and emulsion stability. The findings are expected to provide insights into the development of plant-based, multifunctional emulsifiers that can simultaneously enhance the stability and bioavailability of lipophilic nutrients, supporting their potential applications in functional foods and nutraceuticals.

## 2. Materials and Methods

### 2.1. Materials

Fresh *Lentinula edodes* (shiitake mushrooms) were purchased from a local food market in Fujian, China. Soybean oil was obtained from Shandong Luhua Group Co., Ltd. (Yantai, China) and used without further purification. Caffeic acid, 1-hydroxybenzotriazole (HOBt) and 1-ethyl-3-(3-dimethylaminopropyl) carbodiimide hydrochloride (EDC·HCl), Nile Blue, and Nile Red were procured from Shanghai Aladdin Reagent Co., Ltd. (Shanghai, China). DPPH (2,2-diphenyl-1-picrylhydrazyl), β-carotene, linoleic acid, and Tween 80 were supplied by Shanghai Macklin Biochemical Technology Co., Ltd. (Shanghai, China). For bacterial studies, strains of *Escherichia coli* (*E. coli*) and *Staphylococcus aureus* (*S. aureus*) were sourced from the Microbiological Culture Collection Centre (Guangzhou, China). Other chemicals were acquired from recognized commercial providers.

### 2.2. Extraction and Characterization of Mushroom Chitosan

Mushroom chitosan was extracted from *Lentinula edodes* using an alkaline–acid treatment followed by deacetylation. Fresh mushrooms were first washed, then dried at 50 °C and ground into a fine powder. The powder was treated with 1 M NaOH at 90 °C for 2 h to remove proteins and other impurities, followed by filtration and repeated washing with distilled water until neutral. The residue was then suspended in 2% (*v*/*v*) acetic acid and heated at 80 °C for 3 h, after which the supernatant was collected and precipitated using 95% ethanol to obtain crude chitosan. To achieve a high degree of deacetylation (DDA), the precipitate was refluxed in 50% NaOH at 120 °C for 2 h, followed by extensive washing with distilled water and drying at 60 °C.

The purified chitosan was characterized to determine its physicochemical properties, showing a DDA of 95.6%, a molecular weight (Mw) of 58,362 Da, and good solubility in 1% (*v*/*v*) acetic acid.

### 2.3. Synthesis of Caffeic Acid-Modified Mushroom Chitosan (Chi-g-CA)

Chi-g-CA was synthesized using a previously established method with EDC/HOBt as the coupling agent [11]. Chitosan (1 g) was dissolved in 100 mL of 1% (*v*/*v*) acetic acid aqueous solution. Then, varying amounts of caffeic acid (0.5, 1, and 2 mmol) were added to separate samples. The mixtures were stirred continuously until uniform solutions were obtained, followed by the addition of 2 mmol of HOBt and 2 mmol of EDC·HCl to facilitate conjugation. The reaction mixture was stirred at room temperature in the dark for 24 h. Post-reaction, the grafted chitosan was dialyzed using a dialysis bag (MWCO 8000–14,000) for 48 h to remove unreacted caffeic acid and other small molecules. The resulting product was freeze-dried and stored for further characterization.

Samples were labeled as chitosan grafted with caffeic acid at low (Chi-g-CA L), medium (Chi-g-CA M), and high (Chi-g-CA H) concentrations.

### 2.4. Characterization of Caffeic Acid-Modified Chitosan

UV–Visible spectroscopy (UV-Vis) was performed to analyze both the unmodified chitosan (Chi) and its modified variants. Using a UV-Vis T9 spectrophotometer (Purkinje, Beijing, China), spectral data were collected between 200 nm and 400 nm at 1 nm intervals. These data, in conjunction with a calibration curve from standard caffeic acid solutions, enabled the determination of the grafting degree.

Fourier transform infrared (FTIR) spectroscopy was used to identify the functional groups in Chi and its derivatives. A Nicolet IS 10 FTIR spectrometer (Thermo, Waltham, MA, USA) was employed for this purpose. Samples were prepared using the potassium bromide (KBr) pellet method and scanned over a frequency range of 400 to 4000 cm^−1^.

Proton nuclear magnetic resonance (^1^H NMR) spectroscopy (Bruker, Billerica, MA, USA) was conducted to elucidate the proton changes. The spectra were obtained using a Bruker 400 M ^1^HNMR spectrometer. Chemical shifts were recorded in ppm, referencing the residual solvent signal of D_2_O at δH 4.70 ppm.

X-ray photoelectron spectroscopy (XPS) analysis was carried out using the K-Alpha+ instrument (Thermo, Waltham, USA). The Al Kα ray (HV = 1486.6 eV) was used as the X-ray source, and charge correction was based on the C1s energy standard at 284.80 eV.

### 2.5. Rheological Experiments

The aqueous solutions of Chi and its conjugate Chi-g-CA were prepared at a concentration of 3% *w*/*v*. Rheological properties were determined using a Shbosin RH-20 rheometer (Shanghai Bosin, Shanghai, China) fitted with a 40 mm cone rotor. The viscosity of the solutions was measured across a rotational speed range from 1 to 100 rpm. The impact of temperature on the apparent viscosity of the conjugates was assessed in flow temperature ramp mode, ranging from 20 °C to 80 °C at a rate of 5 °C/min. Dynamic modulus measurements were conducted using a parallel plate geometry (40 mm with a gap of 1 mm) through oscillatory frequency and temperature scans. The linear viscoelastic region (LVR) of the samples was discerned through oscillatory strain sweeps at a consistent frequency of 1 Hz, spanning from 0.1% to 100%. Subsequently, the storage modulus (G′) and loss modulus (G″) of the test sample were determined by a frequency sweep from 1 to 10 Hz, at a fixed strain of 10%. During the temperature scan tests, the sample was heated from 20 °C to 80 °C at a rate of 5 °C/min, and the values of G′ and G″ were recorded against the temperature.

### 2.6. Antioxidant Activity Assay

The antioxidant potential of the chitosan derivative was evaluated using the DPPH radical scavenging assay, following the method [12] with minor modification. A stock solution of DPPH in ethanol (0.2 mM) was prepared. A 50 µL sample of 1% (*w*/*v*) Chi-g-CA was added to the DPPH solution. After a 30 min incubation at 25 °C in darkness, the absorbance was recorded at 517 nm. The percentage of DPPH scavenging was calculated compared to a blank sample containing ethanol instead of the test sample, using Formula (1).(1)$$DPPH\;scavenging\;activity\;(\%)=\frac{A_{control}-A_{sample}}{A_{control}}\times 100\%$$

The β-carotene bleaching assay was performed according to the method described by Rui et al. [13]. A β-carotene/linoleic acid emulsion was prepared by dissolving 0.5 mg of β-carotene in 10 mL of chloroform, 25 μL of linoleic acid, and 200 mg of Tween 80. The chloroform was then removed under vacuum. The residue was immediately diluted with 50 mL of deionized water, followed by vigorous shaking. After adding 0.2 mL of the sample to 5 mL of the β-carotene/linoleic acid emulsion, the absorbance was measured at 470 nm at the start and after 120 min of incubation at 50 °C. Antioxidant activity was calculated based on the rate of β-carotene bleaching relative to the control, using Formula (2).(2)$$Inhibition\;ratio\left(\%\right)=\left(1-\frac{A_{sample}^{0h}-A_{control}^{0h}}{A_{sample}^{2h}-A_{control}^{2h}}\right)\times 100\%$$
where A_0h_ is the initial absorbance of the β-carotene, and A_2h_ is the absorbance after incubation.

### 2.7. Antimicrobial Activity Assay

To evaluate the antimicrobial properties of Chi-g-CA, we adopted the method outlined by Fei et al. [14]. *E. coli* and *S. aureus* were used as test organisms. Briefly, *E. coli* and *S. aureus* were cultured at a density of 1 × 10^6^ CFU/mL and then swabbed uniformly on the surface of agar plates. Wells of 6 mm in diameter were then punched into the agar and filled with 50 µL of the 1% chitosan samples. After incubation at 37 °C for 24 h, the zones of inhibition were measured. A more pronounced zone of inhibition suggests enhanced antimicrobial effectiveness in the sample against the tested microorganism. The experiments were performed in triplicate.

### 2.8. Evaluation of Emulsifying Properties

#### 2.8.1. Emulsion Preparation

For the chitosan-based emulsion, chitosan was solubilized in 1% acetic acid to obtain a 2% *w*/*v* solution. This chitosan solution was then blended with soybean oil in a 1:4 ratio. Emulsification was achieved by homogenizing the mixture at 10,000 rpm for 3 min using a high-shear homogenizer (D-160, DLAB, Shanghai, China).

#### 2.8.2. Droplet Morphology

The type of the emulsion (W/O or O/W) was determined using a dual-staining technique. A 1% solution of Nile Blue and Nile Red was prepared in deionized water. Emulsion samples were mixed with a few drops of these stains and observed under a Leica TCS SP8 laser confocal microscope. Fluorescence in Nile Blue would indicate a water phase, while Nile Red would highlight the oil phase. The emulsion type was identified based on the dispersion pattern of the dyes. This method is adapted from the protocol described by Ribeiro et al. [15]. The droplet morphology was further analyzed by optical microscopy using a CX23 microscope (Olympus Corporation, Tokyo, Japan). Digital images were taken at 200× and analyzed with image analysis software to determine the size distribution and sphericity of the droplets.

#### 2.8.3. Droplet Size and Zeta Potential

The droplet size and zeta potential of the chitosan-based emulsion droplets were determined using a Zetasizer Nano ZS (Malvern Instruments, Malvern, UK). Measurements were performed at 25 °C and pH 4.5. The droplet size was reported as the z-average hydrodynamic diameter, and zeta potential values were obtained using electrophoretic light scattering (ELS).

#### 2.8.4. Emulsifying Activity and Stability

To evaluate the emulsifying activity and stability of modified chitosan, emulsions were prepared with chitosan solution and soybean oil using the method described in Section 2.8.1. The emulsion capabilities were assessed by measuring the emulsion’s turbidity at 500 nm using a UV-Vis spectrophotometer. The emulsifying activity was calculated using Formula (3). Emulsifying stability was evaluated by measuring the change in absorbance after 24 days of standing, using Formula (4).(3)$$Emulsifying\;activity\;(EAI)=\frac{2\times 2.303\times A\times V}{L\times \phi \times C\times 10,000}$$(4)$$Emulsifying\;stability\;\left(ESI\right)=\frac{A_{0}}{A_{0}-A}\times 100\%$$
where A_0_ is the absorbance of the emulsion immediately after homogenization, A is the absorbance at 500 nm, V is the dilution factor, L is the path length of the cuvette (cm), φ is the oil volume fraction of the emulsion, C is the weight of the emulsifier per unit volume of the aqueous phase (g/mL), and 10,000 is a conversion factor to express EAI in m^2^/g.

### 2.9. Encapsulation Efficiency and Bioaccessibility of β-Carotene

#### 2.9.1. Determination of Encapsulation Efficiency

The encapsulation efficiency (EE) of β-carotene in emulsions was determined following a modified method described by a previous study [16]. Emulsions containing β-carotene were centrifuged at 10,000 rpm for 20 min at 4 °C to separate the free β-carotene from the encapsulated β-carotene. The supernatant was collected, and the concentration of free β-carotene was measured using a UV-Vis spectrophotometer (T9, Purkinje General, Beijing, China) at 450 nm. The total β-carotene content in the emulsion was determined by dissolving an aliquot of the emulsion in ethanol under sonication for the complete release of β-carotene. The encapsulation efficiency was calculated using the following formula:(5)$$EE\;\left(\%\right)=\frac{C_{total}-C_{free}}{C_{total}}\times 100\%$$
where *C_total_* means the total β-carotene content in the emulsion (mg/mL), and *C_free_* means the free β-carotene concentration in the supernatant (mg/mL).

#### 2.9.2. Bioaccessibility of β-Carotene

The bioaccessibility of β-carotene in the emulsions was assessed using an in vitro digestion model adapted from a previous study [17]. The model consisted of three sequential digestion phases: oral, gastric, and intestinal.

Oral Phase: The emulsion (10 mL) was mixed with simulated saliva fluid (10 mL, pH 7.0) containing α-amylase at the specific concentration of 75 U/mL. The mixture was incubated at precisely 37 °C for 5 min with gentle agitation (100 rpm) to mimic oral processing.

Gastric Phase: After the oral phase, the pH of the mixture was adjusted to 2.0 ± 0.1 with 1 M HCl, followed by the addition of simulated gastric fluid (10 mL) containing pepsin at the concentration of 2000 U/mL. The resulting solution was incubated at 37 °C for 2 h with continuous gentle shaking (100 rpm) to simulate gastric digestion conditions.

Intestinal Phase: Following the gastric phase, the pH was adjusted to 7.0 ± 0.1 using 1 M NaOH. Simulated intestinal fluid (20 mL) containing pancreatin (100 U/mL) and bile salts (10 mmol/L) was added. This mixture was incubated at 37 °C for 2 h under gentle agitation (100 rpm) to simulate intestinal digestion.

After the complete digestion process, samples were immediately centrifuged at 5000 rpm for 40 min at 4 °C to separate the digested phases. The micellar phase (containing bioaccessible β-carotene) was carefully collected from the middle layer, and its β-carotene content was quantified spectrophotometrically at 450 nm:(6)$$Bioaccessibility\;\left(\%\right)=\frac{C_{micelle}}{C_{disgest}}\times 100\%$$
where C_disgest_ and C_micelle_ are the β-carotene absorbance value in the supernatant fraction and that in the micellar fraction, respectively.

### 2.10. Statistical Analysis

Each experiment was conducted in triplicate. Statistical analysis was performed using one-way ANOVA followed by the independent-samples *t*-test. Differences were considered significant at *p* < 0.05. Results are expressed as mean ± standard deviation (SD), and different letters indicate statistically significant differences between groups.

## 3. Results

### 3.1. UV-Vis Analysis

Figure 1 illustrates the UV-Vis spectra of chitosan modified with varying concentrations of caffeic acid, showcasing the changes induced by the grafting process. Figure 1A highlights a prominent peak around 300 nm, which is observed for both caffeic acid and its chitosan conjugates, a peak that is notably absent in the spectrum of unmodified chitosan. This peak becomes more intense as the concentration of caffeic acid increases, suggesting a higher degree of grafting. The observed increase in peak intensity correlates with the amount of caffeic acid incorporated into the chitosan structure, which is consistent with previous studies, such as the one by Xiao et al. [18], where a similar concentration-dependent increase in grafting efficiency was reported for ferulic acid-grafted arabinoxylan.

Furthermore, the modification of chitosan with caffeic acid not only introduces this distinct peak but also results in an overall shift in the spectral profile, indicating changes in the chemical environment of chitosan. This suggests that the grafting process modifies the chitosan’s structural characteristics. The degree of grafting is further quantified in Figure 1B, where the grafting ratio is shown to increase from 5.02% for Chi-g-CA L to 8.26% for Chi-g-CA H, which demonstrates a clear concentration-dependent trend. This increase can be attributed to the higher availability of active sites for grafting at elevated concentrations of caffeic acid.

### 3.2. FTIR and 1HNMR Analysis

Figure 2 presents the FTIR and 1H NMR spectral analyses of chitosan and its caffeic acid-modified counterparts. Figure 2A,B show the FTIR spectra, where the O-H stretching vibration of the hydroxyl group, a characteristic peak at around 3367 cm^−1^, remains prominent in all samples, confirming that the base structure of chitosan is preserved after modification [19]. In contrast, a new peak at approximately 1636 cm^−1^ appears, corresponding to the carbonyl stretch of caffeic acid, indicating successful grafting. The peaks at 1153 cm^−1^ and 1076 cm^−1^, related to the saccharide structure of chitosan, appear to merge with the C-O-C stretching vibration, further suggesting the incorporation of caffeic acid into the chitosan backbone. Additionally, a distinct peak at 1548 cm^−1^, attributed to the C=C stretching of aromatic rings, becomes more pronounced as the concentration of caffeic acid increases, confirming the presence of phenolic components. The absorption band near 1382 cm^−1^ is likely due to the symmetric stretching of COO- groups from caffeic acid, providing further evidence of the successful grafting of caffeic acid onto chitosan.

Figure 2C displays the 1H NMR spectra of native chitosan and its modified counterparts. A clear differentiation is observed in the aromatic region (6.5–8 ppm), where the peaks corresponding to the caffeic acid protons appear in the modified samples. Figure 2D provides the corresponding chemical shifts for these aromatic protons. These aromatic peaks are absent in the native chitosan spectrum, confirming that no such structures exist in the unmodified polymer. As the concentration of caffeic acid increases from Chi-g-CA L to Chi-g-CA H, these peaks not only become more distinct but also show an increase in intensity, indicating a concentration-dependent grafting of caffeic acid. This chemical shift pattern aligns with the results of both the UV-Vis and FTIR analyses, which suggest a progressive and concentration-dependent modification of chitosan with caffeic acid. Similar findings were reported by Xiao et al. [18], where ferulic acid-grafted arabinoxylan showed comparable aromatic proton peaks in a 1H NMR spectra. Additionally, our previous study on pectin modified with gallic acid demonstrated a similar pattern, where increasing gallic acid concentrations led to more pronounced aromatic proton signals [5].

### 3.3. XPS Analysis

Figure 3 presents the X-ray photoelectron spectroscopy (XPS) analysis of chitosan and its caffeic acid-modified counterparts. The survey spectra in Figure 3A reveal the elemental composition, showing a distinct decrease in the nitrogen atomic percentage from unmodified chitosan (5.22%) to the caffeic acid-grafted variants, with Chi-g-CA L at 4.93%, Chi-g-CA M at 4.74%, and Chi-g-CA H at 4.47%. This reduction in nitrogen content reflects the grafting of caffeic acid, which does not contain nitrogen. Figure 3B further supports this by highlighting the changes in elemental distribution between the unmodified and modified samples. A similar trend of nitrogen content reduction was observed when ferulic acid was grafted onto chitosan [20].

In Figure 3C, the C1s spectra demonstrate a notable change in peak intensity at 284.92 eV for the modified chitosans, indicating an increased presence of carbon–nitrogen bonds, characteristic of the grafted phenolic groups. This shift further supports the successful incorporation of caffeic acid into the chitosan structure. Additionally, Figure 3D presents the N1s spectra, providing conclusive evidence of grafting. A new peak at approximately 401.7 eV appears in the modified chitosan samples, corresponding to the formation of acylamino groups, which are indicative of the amide bonds formed between chitosan and caffeic acid. This finding is in line with the study by Fei et al. [14], who reported a similar peak in the N1s spectra of chitosan modified with p-coumaric acid, confirming that the formation of acylamino groups is a common feature in chitosan modified with phenolic acids.

### 3.4. Rheological Properties

Figure 4 presents the rheological behavior of chitosan after modification with caffeic acid, revealing important relationships between structural changes and functional properties. The analysis of the viscosity, storage modulus (G′), and loss modulus (G″) provides valuable insights into how caffeic acid grafting influences the material’s behavior in solution and its potential performance as an emulsifier. Figure 4A illustrates that all chitosan samples exhibit pseudoplastic (shear-thinning) behavior. Notably, the chitosan–caffeic acid derivatives show lower viscosity than unmodified chitosan across all rotational speeds, which aligns with findings by Wang et al. [21] for ferulic acid-grafted carboxylic curdlan conjugates. This decrease in viscosity can be attributed to chain scission during the EDC/HOBt-mediated grafting process, where the hydrolysis of chitosan chains may occur, reducing their length and resulting in lower viscosity. However, an important observation is that as the concentration of caffeic acid increases, a slight viscosity increase is observed among the modified samples, suggesting that the additional phenolic groups enhance intermolecular interactions, forming a more entangled network despite the shortened chain length [22]. The temperature-dependent viscosity profiles in Figure 4B demonstrate that all chitosan solutions show typical polymer behavior, with the viscosity decreasing as the temperature rises. Significantly, the samples with higher caffeic acid grafting show a more gradual viscosity reduction with temperature increase. This thermal behavior suggests that the modified chitosan forms a more structurally resilient network that better maintains its integrity under thermal stress. The phenolic structures likely create additional junction zones through hydrogen bonding and hydrophobic interactions that resist thermal disruption.

Frequency sweep analysis (Figure 4C,D) reveals critical information about the viscoelastic nature of these materials. The storage modulus (G′) of all samples decreases with increasing frequency, characteristic of relaxation phenomena in polymeric networks. The modified chitosans consistently demonstrate higher G′ values compared to unmodified chitosan, indicating enhanced elastic response. This improvement in elasticity despite reduced viscosity creates a particularly advantageous rheological profile for emulsion stabilization, where elasticity contributes significantly to long-term emulsion stability by resisting droplet coalescence. The loss modulus (G″) profiles in Figure 4D show relatively consistent values across the frequency sweep for all chitosan variants, indicating that the energy dissipation mechanisms within the modified chitosan structure remain relatively stable under varying oscillatory stress. The comparative patterns of G′ and G″ suggest that at lower frequencies, elastic behavior dominates, while at higher frequencies, a crossover toward more viscous characteristics occurs.

The temperature sweep rheology (Figure 4E,F) adds to our understanding of these materials’ thermal response. Both G′ and G″ decrease for all samples with increasing temperature, attributable to increased molecular mobility that reduces the material’s ability to store elastic energy. Despite this general trend, the modified chitosan samples, especially Chi-g-CA H, maintain higher moduli throughout the temperature range, demonstrating the superior thermal stability of their network structure.

This comprehensive rheological characterization reveals that caffeic acid grafting transforms chitosan’s solution properties in ways that optimize its performance as an emulsifier. The moderate viscosity facilitates efficient homogenization during emulsion preparation, while the enhanced elastic network provides the structural integrity necessary for stabilizing the oil–water interface over time. These rheological characteristics help explain the improved emulsification performance observed in subsequent experiments.

### 3.5. Antioxidant and Antimicrobial Activity

The antioxidant and antimicrobial properties of chitosan play a crucial role in the effectiveness of emulsions, particularly in applications such as food preservation, pharmaceuticals, and cosmetics, where protection from oxidative degradation and microbial contamination is essential. Figure 5 illustrates the enhancement of these properties in chitosan derivatives as a result of caffeic acid grafting. Figure 5A demonstrates a significant increase in DPPH radical scavenging activity with higher concentrations of caffeic acid, highlighting the role of caffeic acid in improving the antioxidant capacity of chitosan. The unmodified chitosan shows minimal activity (0.39%), which is dramatically elevated in Chi-g-CA H (45.75%). This increase can be attributed to the phenolic hydroxyl groups in caffeic acid, which are highly effective in electron donation and stabilizing free radicals [8].

Figure 5B presents a similar trend in the β-carotene bleaching assay, where the chitosan–caffeic acid derivatives show increased inhibition percentages, with Chi-g-CA H reaching up to 44.82%. This aligns with studies by Rui et al. [13] and Yi et al. [23], who observed enhanced antioxidant activity in chitosan derivatives with higher degrees of substitution in similar assays. The consistency of results across different assays strengthens the conclusion that caffeic acid significantly enhances the antioxidant capacity of chitosan, thereby improving the stability and shelf life of emulsions.

Figure 5C–E demonstrate the antimicrobial efficacy of chitosan and its caffeic acid-modified variants against *Escherichia coli* and *Staphylococcus aureus*. A noticeable increase in the zones of inhibition is observed for both bacterial strains. Specifically, the inhibition zone against *E. coli* increases from 1.27 cm for unmodified chitosan to 1.58 cm for Chi-g-CA H, and for *S. aureus*, the zone of inhibition increases from 1.75 cm to 2.14 cm. These improvements suggest that the grafting of caffeic acid enhances the antimicrobial potency of chitosan, which could be attributed to a synergistic effect between chitosan and caffeic acid. This finding indicates that the grafting process not only retains, but also amplifies, the inherent antimicrobial properties of both chitosan and caffeic acid.

Similar observations have been reported regarding the enhanced antimicrobial effects of chitosan derivatives against various bacterial strains, further supporting our findings [21]. The enhanced antimicrobial activity of Chi-g-CA can be attributed to the combined effects of chitosan’s ability to disrupt microbial cell walls and the antimicrobial properties of caffeic acid. As noted by Yang et al. [24], the structural disruption of microbial cell walls by chitosan, enhanced by phenolic compounds, is crucial in the observed improvement in antimicrobial efficacy.

### 3.6. Emulsion Droplet Morphology

In Figure 6, selective dyes are used to reveal the O/W emulsion structure, with Nile Blue staining the continuous water phase in blue and Nile Red highlighting the dispersed oil droplets in red. The clear blue background with discrete red droplets confirms that the oil droplets are encapsulated by the water phase, characteristic of an O/W emulsion. As the amount of caffeic acid grafted onto chitosan increases, the droplet size decreases and becomes more uniform. This observation aligns with the findings of Luan et al. [25], who reported that Arabic gum grafted with phenolic acid, similarly to our modified chitosan, significantly enhanced the stability of O/W emulsions.

The optical images on the right further quantify this observation, showing a decrease in the mean droplet size from native chitosan (Chi) to Chi-g-CA H emulsions. This significant reduction in droplet size with increased caffeic acid concentration suggests a more efficient emulsification process, likely due to the enhanced amphiphilic properties of the modified chitosan. These properties improve the stabilization of the interface between the oil and water phases, facilitating better emulsion formation.

### 3.7. Emulsifying Properties

Building on the analysis of emulsion droplet morphology in the previous section, where changes in droplet size and uniformity were observed with increasing caffeic acid grafting, the emulsifying properties of chitosan and its caffeic acid-modified counterparts are further explored. The emulsifying capacity, stability over time, ζ-potential, and droplet size distribution of the emulsions are critical factors influencing emulsion performance in various applications. Figure 7 presents these properties, showcasing the positive effects of caffeic acid grafting on chitosan’s emulsifying performance.

The overall improvement in emulsifying properties indicates that the modification of chitosan with caffeic acid not only enhances its emulsifying capability but also contributes to the stability and efficiency of the resulting emulsions. As shown in Figure 7A, the emulsifying activity increases significantly from 1.95 units for unmodified chitosan to 3.32 units for Chi-g-CA H, suggesting that caffeic acid enhances chitosan’s amphiphilic nature, which facilitates better emulsion formation [26].

Figure 7B demonstrates that all emulsion systems exhibited progressive destabilization over the 24-day observation period, reflecting the inherent thermodynamic instability characteristic of oil-in-water emulsions. However, emulsions with higher caffeic acid content showed significantly slower rates of destabilization. This can be attributed to multiple mechanisms: coalescence appears to dominate in unmodified chitosan emulsions, as evidenced by a rapid stability decline, while Chi-g-CA H emulsions exhibit enhanced resistance to both coalescence and Ostwald ripening. The improved stability likely stems from the enhanced amphiphilicity and interfacial properties of caffeic acid-modified chitosan, creating more robust interfacial films that resist rupture during droplet collision [27,28]. These findings align with previous studies, where Luan et al. [25] demonstrated that Arabic gum grafted with phenolic acids enhanced oil-in-water emulsion stability through improved interfacial properties, and Cheng et al. [29] reported that caffeic acid complexation with proteins improved dispersion stability via hydrophobic interactions. This study extends these principles to chitosan systems, though it should be noted that complete thermodynamic stability remains unattainable in conventional emulsion systems.

In terms of ζ-potential, Figure 7C shows that the grafting of caffeic acid leads to an increase in ζ-potential from 32.23 mV for unmodified chitosan to 55.63 mV for Chi-g-CA H, which enhances the electrostatic repulsion between emulsion droplets [30]. This increased electrostatic repulsion contributes to better stability by preventing droplet coalescence [31]. Finally, Figure 7D shows a reduction in droplet size from 2790 nm for unmodified chitosan to 1253.33 nm for Chi-g-CA H, further supporting the improved emulsification efficiency and stability due to caffeic acid grafting. The smaller droplet size suggests more efficient emulsification, likely attributed to the denser packing of modified chitosan at the oil–water interface, creating a more effective barrier against droplet fusion. In detail, caffeic acid grafting introduces both hydrophilic (hydroxyl groups) and hydrophobic (aromatic rings) moieties to the chitosan backbone, enhancing its amphiphilic character. This structural modification allows the caffeic acid-grafted chitosan to orientate more effectively at the oil–water interface, with hydrophobic segments extending into the oil phase and hydrophilic portions remaining in the aqueous phase. The phenolic structures of caffeic acid likely facilitate π-π stacking interactions between adjacent molecules, promoting closer molecular arrangement and consequently denser packing at the interface. Additionally, the increased zeta potential observed in modified chitosan may enhance electrostatic repulsion between adsorbed polymer chains, leading to the more uniform coverage of the interface. This combination of enhanced amphiphilicity and intermolecular interactions results in a more robust interfacial film that effectively prevents droplet coalescence.

In conclusion, the modification of chitosan with caffeic acid significantly improves its emulsifying properties, resulting in emulsions with enhanced stability, reduced droplet size, and increased emulsifying activity. These improvements are likely due to the increasingly amphiphilic nature of the modified chitosan, which enhances both the formation and the stability of the emulsion. This underscores the potential of Chi-g-CA as a more effective emulsifier for a wide range of applications, from food science to pharmaceuticals.

### 3.8. Delivery of β-Carotene

The encapsulation efficiency of β-carotene in emulsions has garnered attention for its potential health benefits despite its hydrophobic nature and limited bioavailability. Figure 8 presents the encapsulation efficiency and bioaccessibility of β-carotene in chitosan-based emulsions, showcasing the impact of caffeic acid modification on both parameters. Figure 8A shows the encapsulation efficiency, while Figure 8B illustrates the bioaccessibility of β-carotene, both of which are critical in evaluating the effectiveness of emulsions in enhancing the bioavailability of β-carotene.

As shown in Figure 8A, modified chitosan emulsions demonstrated superior encapsulation efficiency compared to native chitosan. Specifically, Chi-g-CA H exhibited the highest encapsulation efficiency (87.46%), followed by Chi-g-CA M (82.34%), Chi-g-CA L (75.31%), and native chitosan (65.28%). The enhanced encapsulation efficiency of Chi-g-CA emulsions can be attributed to multiple factors, including the increased hydrophobicity and improved amphiphilic balance imparted by caffeic acid grafting. This modification strengthens the interactions between β-carotene and the hydrophobic domains of the emulsifier, facilitating more efficient entrapment within the emulsion droplets. Furthermore, the denser packing of modified chitosan at the oil–water interface, as discussed earlier, creates a more robust physical barrier that effectively prevents the migration of β-carotene molecules from the oil phase to the aqueous phase. This enhanced interfacial structure not only improves emulsion stability but also significantly contributes to the higher retention of β-carotene within the oil droplets, resulting in the observed increase in encapsulation efficiency from 65.28% for unmodified chitosan to 87.46% for Chi-g-CA H. These findings align with previous studies, underscoring the importance of hydrophobic interactions and structural compatibility in improving encapsulation efficiency [32,33].

The bioaccessibility of β-carotene was assessed to evaluate the effectiveness of modified chitosan in enhancing nutrient delivery. Figure 8B shows that β-carotene bioaccessibility is significantly higher in Chi-g-CA emulsions compared to native chitosan. Chi-g-CA H demonstrated the highest bioaccessibility (52.13%), followed by Chi-g-CA M (47.89%), Chi-g-CA L (40.26%), and native chitosan (31.56%). The improved bioaccessibility can be attributed to the reduced droplet size and higher zeta potential of the modified chitosan emulsions, which promote greater stability during digestion and better micelle formation. The smaller droplet size observed in Chi-g-CA H emulsions likely facilitated the digestion process by increasing the surface area available for enzymatic action, while the higher zeta potential maintained emulsion stability during digestion. These properties collectively enhanced the formation of mixed micelles with bile salts in the intestinal phase, increasing the solubilization of β-carotene and its subsequent incorporation into the micellar phase available for absorption. These properties collectively enhanced the solubilization and bioavailability of β-carotene in the intestinal environment.

## 4. Discussion

This study demonstrates the development of a novel soybean oil-based emulsion system using Chi-g-CA, significantly enhancing the functional properties of chitosan and its application in the delivery of β-carotene.

### 4.1. Impact of Caffeic Acid Modification on Chitosan Structure and Function

UV-Vis, FTIR, NMR, and XPS analyses confirmed the successful grafting of caffeic acid onto the chitosan backbone via an EDC/HOBt-mediated reaction. The grafting degree increased from 5.02% to 8.26% as the concentration of caffeic acid increased (Figure 1, Figure 2 and Figure 3). This chemical modification enhanced the amphiphilic properties of chitosan, with the incorporation of caffeic acid’s phenolic ring (noted from the characteristic peak at 6.5–8 ppm in NMR) significantly strengthening chitosan’s interaction with hydrophobic substances. This was reflected in the improved encapsulation efficiency of β-carotene (Figure 8A). Additionally, the phenolic hydroxyl groups of caffeic acid contributed to the system’s enhanced antioxidant activity through radical scavenging (Figure 5A,B). Furthermore, the synergistic interaction between the positive charges of chitosan and the negative carboxyl groups (COO⁻) of caffeic acid improved antimicrobial performance by disrupting microbial cell membranes (Figure 5C–F).

It is noteworthy that although the viscosity of the modified chitosan decreased after grafting (Figure 4A,B), which may be attributed to the partial hydrolysis of the chitosan backbone during the EDC/HOBt reaction, the significant increase in the elastic modulus (G′) (Figure 4C) indicates the formation of a new crosslinked network structure. This structural change plays a crucial role in the stability of the subsequent emulsion.

### 4.2. Multiple Mechanisms Behind Emulsion Stability

The Chi-g-CA-based emulsion system exhibited superior initial performance characteristics, including smaller droplet sizes (1253 nm vs. 2790 nm) and higher zeta potential (55.63 mV vs. 32.23 mV), as shown in Figure 6 and Figure 7, which are predictive indicators of potential improved stability. The enhanced stability can be attributed to several synergistic mechanisms. First, the amphiphilic structure of the modified chitosan, with the hydrophobic caffeic acid segment preferentially anchoring at the oil droplet surface and the hydrophilic chitosan backbone extending into the aqueous phase, effectively forms a steric barrier. Additionally, the electrostatic repulsion between the cationic chitosan and the anionic caffeic acid groups (e.g., COO⁻) significantly enhanced the electrostatic repulsion between droplets (Figure 7C), effectively preventing droplet aggregation.

Another key stability mechanism arises from the unique rheological properties of the Chi-g-CA solution, with its higher elastic modulus (G′ > G″) (Figure 4C,D) indicating the formation of a gel-like network structure, which effectively restricts droplet migration. Moreover, the emulsion stability showed a clear positive correlation with the grafting degree of caffeic acid (Figure 7A,B), further confirming the decisive role of the amphiphilic modification in emulsification performance and providing valuable theoretical insights for the design and optimization of natural emulsifiers.

### 4.3. Mechanism Behind Improved β-Carotene Delivery Efficiency

The Chi-g-CA emulsion system exhibited a remarkable enhancement in β-carotene encapsulation efficiency (87.46%) and bioavailability (52.13%), significantly outperforming the unmodified system (Figure 8). This performance improvement can be explained on three levels: First, the hydrophobic phenyl ring of caffeic acid creates a favorable hydrophobic microenvironment, enhancing the entrapment of β-carotene through π-π stacking and hydrophobic interactions (Figure 8A). Second, the high zeta potential and smaller droplet size (Figure 7C,D) of the emulsion significantly reduced droplet aggregation during digestion, while the antioxidant activity of Chi-g-CA effectively delayed lipid oxidation (Figure 5A,B), providing “dual protection” for β-carotene. Third, the improved physicochemical properties of Chi-g-CA emulsions, particularly the smaller droplet size and enhanced stability during digestion conditions, may have facilitated the release of β-carotene into the aqueous phase of the digestive medium [34,35], as reflected in the improved bioaccessibility results shown in Figure 8B. This enhancement is likely due to the improved interfacial activity resulting from its optimized amphiphilic structure.

### 4.4. Limitations and Future Perspectives

Despite demonstrating the potential of Chi-g-CA in food emulsifier applications, several issues warrant further attention. First, the long-term stability of the emulsion system, especially under storage conditions involving light, temperature, and oxidation, needs to be systematically evaluated. Second, the introduction of caffeic acid may affect the color and flavor of the emulsion, which should be optimized through sensory evaluation experiments. Furthermore, the current bioavailability data are primarily based on in vitro simulated digestion, and further verification through animal studies is necessary.

## 5. Conclusions

In this study, a novel soybean oil-based emulsion system was developed using Chi-g-CA, which significantly enhanced the functional properties of chitosan, including its emulsifying, antioxidant, antimicrobial, and delivery capabilities. The structural modification of chitosan with caffeic acid improved its amphiphilic nature, facilitating better interactions with hydrophobic substances and enhancing β-carotene encapsulation efficiency. The emulsion system exhibited improved stability, characterized by smaller droplet sizes, higher zeta potential, and superior long-term stability, which can be attributed to the synergistic effects of electrostatic repulsion, steric barriers, and gel-like network structures.

Furthermore, the Chi-g-CA emulsion demonstrated a significant improvement in β-carotene delivery efficiency, enhancing both encapsulation and bioavailability, which was attributed to the optimized amphiphilic structure and the protective effects against lipid oxidation and aggregation during digestion. The superior emulsifying properties of Chi-g-CA, including smaller droplet size and enhanced interfacial stability during simulated digestion, contributed to the observed improvement in β-carotene bioaccessibility.

While this study highlights the potential of Chi-g-CA as a multifunctional food emulsifier, further research is required to address the emulsion’s stability under various environmental conditions, optimize sensory properties, and validate bioavailability through in vivo studies. Overall, this work provides a promising approach to the development of natural multifunctional emulsifiers, with significant potential for applications in functional foods and nutrient delivery systems.

## Figures and Tables

**Figure 1 foods-14-01108-f001:**
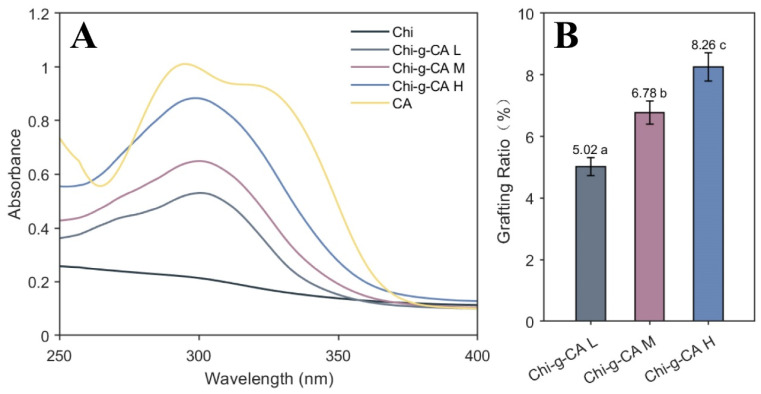
UV-Vis spectra (**A**) and grafting ratios (**B**) for chitosan and its derivatives. Different letters indicate statistically significant differences between groups at *p* < 0.05.

**Figure 2 foods-14-01108-f002:**
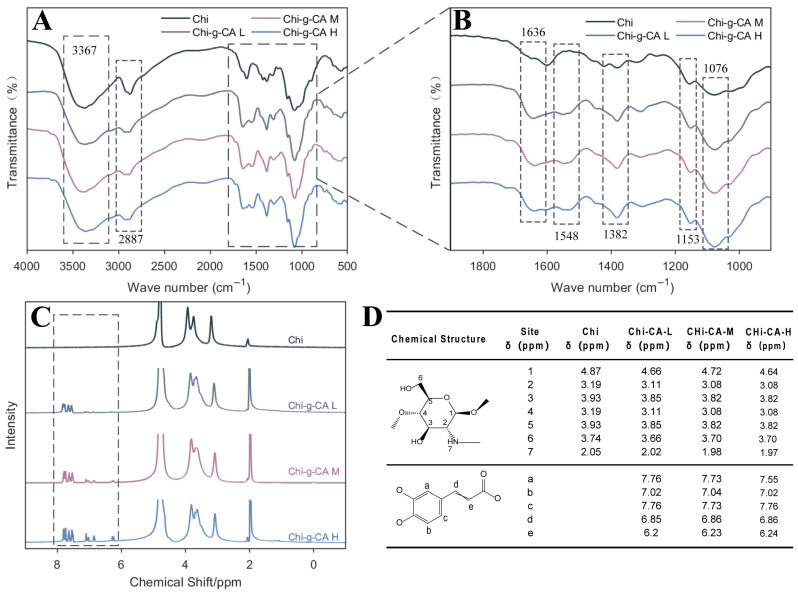
FTIR and ^1^H NMR spectra of chitosan and its derivatives. (**A**,**B**) FTIR spectra with characteristic transmittance peaks; (**C**) the 1H NMR spectra with resonance signals. (**D**) Table of chemical shifts for each sample.

**Figure 3 foods-14-01108-f003:**
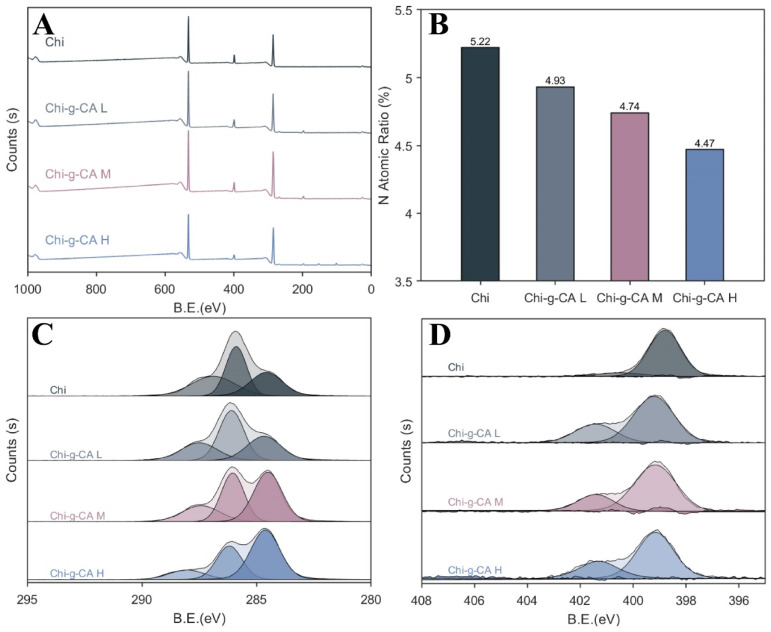
XPS spectra of chitosan and its derivatives. (**A**) The full survey spectra; (**B**) the nitrogen atomic percentage in each sample; (**C**) the high-resolution C1s spectra; (**D**) the high-resolution N1s spectra.

**Figure 4 foods-14-01108-f004:**
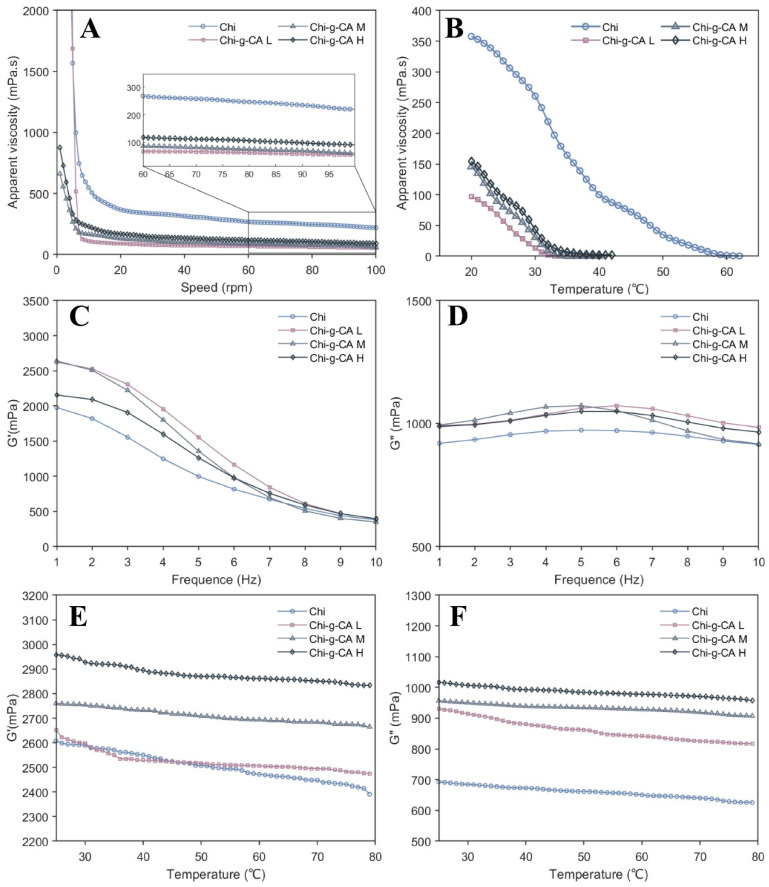
Rheological properties of chitosan and its derivatives. (**A**) Apparent viscosity versus shear rate for chitosan samples. (**B**) Effect of temperature on viscosity. (**C**,**D**) Frequency sweeps of the storage modulus (G′) and loss modulus (G″); (**E**,**F**) temperature sweeps of G′ and G″.

**Figure 5 foods-14-01108-f005:**
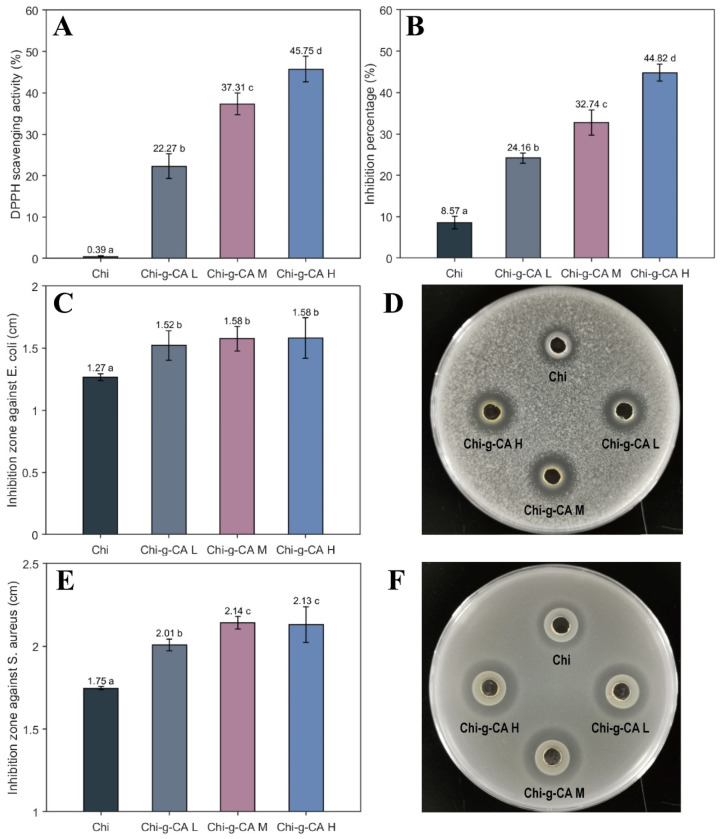
Antioxidant and antimicrobial activity of chitosan and its derivatives. (**A**) DPPH radical scavenging activity; (**B**) inhibition percentages in the β-carotene bleaching assay; (**C**,**D**) inhibition zones and representative image against *E. coli*; (**E**,**F**) inhibition zones and representative image against *S. aureus.* Different letters indicate statistically significant differences between groups at *p* < 0.05.

**Figure 6 foods-14-01108-f006:**
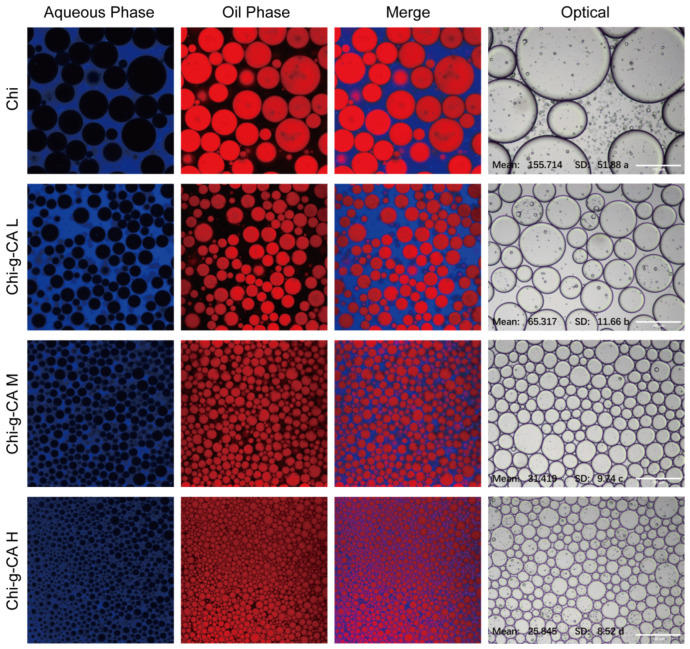
Confocal and optical microscopy images of chitosan and its derivatives.

**Figure 7 foods-14-01108-f007:**
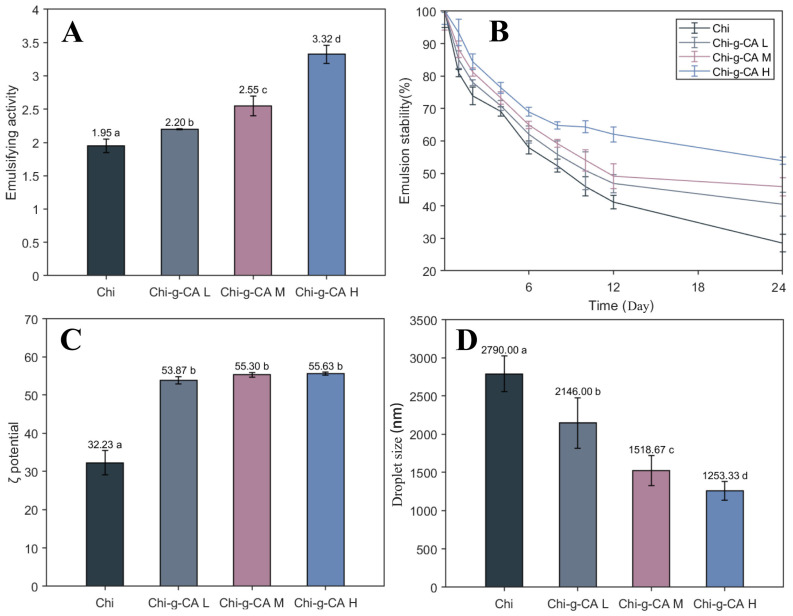
Emulsifying capacity of chitosan and its derivative emulsions. (**A**) Emulsifying activity; (**B**) time-dependent stability; (**C**) ζ-potential values; (**D**) droplet sizes. Different letters indicate statistically significant differences between groups at *p* < 0.05.

**Figure 8 foods-14-01108-f008:**
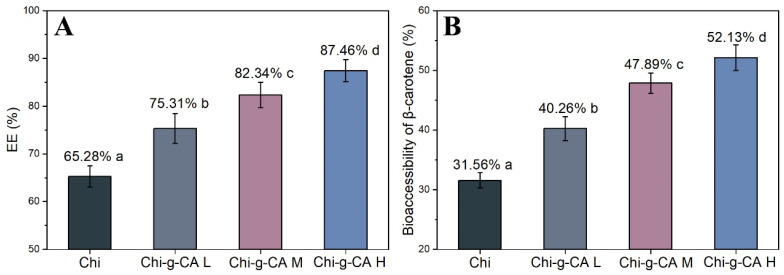
Encapsulation efficiency (**A**) and bioaccessibility (**B**) of β-carotene in emulsions stabilized by chitosan and its derivatives. Different letters indicate statistically significant differences between groups at *p* < 0.05.

## Data Availability

The original contributions presented in the study are included in the article; further inquiries can be directed to the corresponding author.

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
