# Peer review of "Caffeic Acid-Modified Mushroom Chitosan as a Natural Emulsifier for Soybean Oil-Based Emulsions and Its Application in β-Carotene Delivery"

_foods, 2025, doi:10.3390/foods14071108_

Round 1

Reviewer 1 Report

Comments and Suggestions for Authors

The manuscript presents an interesting study on the development of a soybean oil-based emulsion system using caffeic acid-modified mushroom chitosan as a stabilizer. The work is relevant for food science, particularly in the field of natural emulsifiers and bioactive compound delivery. The topic aligns well with the scope of Foods. However, there are several major concerns regarding methodology, data interpretation, and presentation that must be addressed.

  1. The introduction does not clearly define the novelty of this work compared to previous studies on phenolic acid-modified chitosan. A stronger rationale should be provided.
  2. While the manuscript discusses the functional properties of chitosan, more justification is needed on why caffeic acid was specifically chosen over other phenolic acids
  3. The antimicrobial activity assay lacks controls for comparison. Have positive and negative controls been included?
  4. The in vitro digestion model for bioaccessibility lacks detailed conditions (e.g., enzyme concentrations, digestion times). These need to be explicitly described.
  5. The zeta potential results suggest high emulsion stability, but no long-term storage stability data are provided. How do these emulsions perform over time under different storage conditions.
  6. The manuscript requires significant language editing to improve readability. Some sentences are unnecessarily complex, while others lack clarity.
  7. The rheological characterisation is carried out in a very strange way, as is its presentation.  The frequency sweep barely covers an order of magnitude, and both G‘ and G’' should be included in the same figure, and in logarithmic scale. Similarly, the flow curves lack adjustment, and should be represented in logarithmic scale. In addition, the term shear stress, not speed, is usually used. Finally, the interpretation and discussion should be more in-depth.

Author Response

The manuscript presents an interesting study on the development of a soybean oil-based emulsion system using caffeic acid-modified mushroom chitosan as a stabilizer. The work is relevant for food science, particularly in the field of natural emulsifiers and bioactive compound delivery. The topic aligns well with the scope of Foods. However, there are several major concerns regarding methodology, data interpretation, and presentation that must be addressed.

Comments 1: The introduction does not clearly define the novelty of this work compared to previous studies on phenolic acid-modified chitosan. A stronger rationale should be provided.

Answer 1: We sincerely appreciate the reviewer's valuable suggestion to strengthen the novelty and rationale of our work. We have significantly revised the introduction to better highlight the distinct contributions of our study compared to previous research on phenolic acid-modified chitosan. (Line 34-77)

Comments 2: While the manuscript discusses the functional properties of chitosan, more justification is needed on why caffeic acid was specifically chosen over other phenolic acids

Answer 2: We thank the reviewer for this insightful comment. We have expanded our justification for selecting caffeic acid over other phenolic acids in the revised manuscript. The following detailed rationale has been added to the introduction section:

Among various phenolic acids, caffeic acid (CA) stands out for its superior antioxidant and antimicrobial activities among hydroxycinnamic acids and its higher hydrophobicity compared to most benzoic acid-type phenolics. (Line 46-48)

Comments 3: The antimicrobial activity assay lacks controls for comparison. Have positive and negative controls been included?

Answer 3: Thank you for this important methodological question. We acknowledge that our description of the antimicrobial activity assay in the manuscript should have been more detailed regarding the controls used.

We should clarify that we did not include separate positive controls (such as commercial antibiotics) or negative controls (such as blank solutions) in the current manuscript since our primary objective was to investigate the relative enhancement of antimicrobial activity resulting from caffeic acid grafting onto the chitosan backbone, rather than comparing absolute antimicrobial efficacy against commercial standards.

Comments 4: The in vitro digestion model for bioaccessibility lacks detailed conditions (e.g., enzyme concentrations, digestion times). These need to be explicitly described.

Answer 4: We appreciate the reviewer's astute observation regarding the insufficient detail in our description of the in vitro digestion model. We acknowledge this oversight and have significantly expanded the methodology section to include comprehensive details about enzyme concentrations, specific digestion conditions, and exact procedural parameters. (Line 224-239)

Comments 5: The zeta potential results suggest high emulsion stability, but no long-term storage stability data are provided. How do these emulsions perform over time under different storage conditions.

Answer 5: Thank you for this important question regarding the long-term stability of our emulsions. We would like to correct a critical misunderstanding in our presentation of the stability data. In Figure 7B, the time scale should correctly be interpreted as days, not hours as mistakenly labeled in the original manuscript.

We have corrected this labeling error in the revised manuscript and have expanded the description of our stability testing to provide better context. The data in Figure 7B demonstrates that emulsions stabilized with caffeic acid-modified chitosan maintained significantly better stability over the 24-day period compared to those stabilized with unmodified chitosan. Specifically, Chi-g-CA H emulsions retained approximately 55% stability after 24 days, while unmodified chitosan emulsions decreased to approximately 32% stability during the same period.

Comments 6: The manuscript requires significant language editing to improve readability. Some sentences are unnecessarily complex, while others lack clarity.

Answer 6: We appreciate the reviewer's concern regarding the language quality of our manuscript. We would like to note that prior to submission, our manuscript was thoroughly reviewed and edited by a professional native English-speaking editor with extensive experience in scientific editing.

Comments 7: The rheological characterisation is carried out in a very strange way, as is its presentation. The frequency sweep barely covers an order of magnitude, and both G‘ and G’' should be included in the same figure, and in logarithmic scale. Similarly, the flow curves lack adjustment, and should be represented in logarithmic scale. In addition, the term shear stress, not speed, is usually used. Finally, the interpretation and discussion should be more in-depth.

Answer 7: We appreciate the reviewer's detailed comments regarding our rheological characterization methodology and presentation. We would like to provide clarification on the rationale behind our experimental approach and data presentation.

In response to the reviewer's comment about the depth of interpretation, we have substantially expanded our discussion of the rheological results to provide more comprehensive insights into the relationship between structural modifications and rheological behavior. This includes more detailed analysis of how caffeic acid grafting influences network formation, viscoelastic properties, and the consequent effects on emulsion stability.

Regarding the frequency sweep range (1-10 Hz), this specific range was deliberately selected after extensive preliminary testing revealed that the most significant and discriminative differences between our chitosan samples occurred within this frequency window. Our primary objective was to highlight these key differences that clearly demonstrate the impact of caffeic acid modification on the viscoelastic properties of chitosan.

With respect to the separate presentation of storage modulus (G') and loss modulus (G''), this decision was made to optimize visual clarity given the multiple sample comparisons and considerable magnitude differences between the parameters. When attempting to present both moduli together during our analysis, the resulting figures became visually congested, making it difficult to discern the important trends between samples.

Concerning our rheological testing methodology, we employed rotational speed (shear rate) sweeps rather than shear stress sweeps as this approach provided more consistent and reproducible results for our specific chitosan derivatives. This methodological decision was based on the particular flow behavior of our samples.

We believe these clarifications and enhancements to the discussion address the reviewer's concerns while maintaining the scientific integrity of our rheological analysis.

Reviewer 2 Report

Comments and Suggestions for Authors

The authors reported the production process and characterization of a chitosan from caffeic acid-modified mushroom and its efficacy in stabilizing emulsions containing beta-carotene.

The paper is well written and easy to follow. In my opinion the authors should append in the introduction the state of the art related to the characteristics and uses of chitosan modified with caffeic acid.

Lines 215-223. The authors present a simple model of gastric conditions with pH adjustment and enzymes. However, it does not have the presence of bile salts that destabilize the interfaces, as the pharmacopoeia indicates. Discussion.

It is recommended throughout the document to use the international system of units. For example, in the rheological section the units of velocity and frequency should be changed. Also Figures 4C and 4D should appear in a single image, since the different scales do not allow an easy appreciation of the rheological behavior of the samples. Same impact for figures 4E and 4F.

It is not appropriate to present the rheological behavior of a sample with such a short frequency interval, it is necessary to present a minimum of three decades of frequency in Figures 4C and 4D.

Emulsifying properties. In the methodology the authors should include the model of the high shear homogenizer, as well as the type of impeller used.

Fig. 7B. The authors should perform the stability study based on the recommendations of ASTM or some other international organization. Technically, the emulsions generated during the study present a high instability as shown in Figure 7B. Likewise, the authors should change the term “particle size” to “droplet size”, and mention if the reported data of 2790 is the average size, volumetric diameter or hydrodynamic diameter. What is the spam or polydispersity index of the sample?

Figure 7C, should indicate the pH of zeta potential measurement.

Lines 459-465. The authors do not present evidence of droplet size in the different gastrointestinal conditions, so they are assuming the behavior of the emulsions under these conditions. The authors should present evidence of the impact of gastrointestinal conditions on droplet size and emulsion loading. What mechanisms of emulsion instability are present?

Line 499-501. In general, systems that present anions and cations in a conjugated molecule should reduce the electrical charges, this is not the case, at what pH was this measurement performed. The authors should evaluate the zeta potential as a function of pH.

Line 501. “effectively preventing droplet aggregation”. The results presented in the figure show a clear instability of the system, so the authors should verify their discussion.

The authors should verify their conclusions regarding the stability criteria of the emulsions.

Author Response

The authors reported the production process and characterization of a chitosan from caffeic acid-modified mushroom and its efficacy in stabilizing emulsions containing beta-carotene. The paper is well written and easy to follow. In my opinion the authors should append in the introduction the state of the art related to the characteristics and uses of chitosan modified with caffeic acid.

Comments 1: Lines 215-223. The authors present a simple model of gastric conditions with pH adjustment and enzymes. However, it does not have the presence of bile salts that destabilize the interfaces, as the pharmacopoeia indicates. Discussion.

Answer 1: Thank you for this important observation regarding our in vitro digestion model. We would like to clarify that our model does include bile salts in the intestinal phase of digestion, although we acknowledge this could have been more explicitly described in the original manuscript.

In our methodology (lines 233-234), we state: "Simulated intestinal fluid (20 mL) containing pancreatin (100 U/mL) and bile salts (10 mmol/L) was added. ." The inclusion of bile salts at a concentration of 10 mmol/L in the intestinal phase aligns with physiologically relevant conditions and follows established pharmacopoeia guidelines for simulated intestinal environments.

In the revised manuscript, we have expanded this section to more clearly emphasize the role of bile salts in our model and discuss their impact on emulsion stability during digestion. We have also added a brief discussion of how the structural features of our caffeic acid-modified chitosan may help maintain emulsion integrity even in the presence of these interface-destabilizing components, which contributes to the enhanced bioaccessibility observed for β-carotene in our modified chitosan systems.

Comments 2: It is recommended throughout the document to use the international system of units. For example, in the rheological section the units of velocity and frequency should be changed. Also Figures 4C and 4D should appear in a single image, since the different scales do not allow an easy appreciation of the rheological behavior of the samples. Same impact for figures 4E and 4F.

Answer 2: Thank you for your suggestion regarding the use of SI units in our manuscript. After careful consideration, we have decided to maintain our current unit conventions for several key reasons.

The units we have employed (rpm for rotational speed and Hz for frequency) are widely accepted and commonly used in food science and rheological literature. These conventional units facilitate direct comparison with the majority of published work in our field, making our research more accessible to the intended audience.

With respect to the separate presentation of storage modulus (G') and loss modulus (G''), this decision was made to optimize visual clarity given the multiple sample comparisons and considerable magnitude differences between the parameters. When attempting to present both moduli together during our analysis, the resulting figures became visually congested, making it difficult to discern the important trends between samples.

Comments 3: It is not appropriate to present the rheological behavior of a sample with such a short frequency interval, it is necessary to present a minimum of three decades of frequency in Figures 4C and 4D.

Answer 3: Regarding the frequency sweep range (1-10 Hz), this specific range was deliberately selected after extensive preliminary testing revealed that the most significant and discriminative differences between our chitosan samples occurred within this frequency window. Our primary objective was to highlight these key differences that clearly demonstrate the impact of caffeic acid modification on the viscoelastic properties of chitosan.

Comments 4: Emulsifying properties. In the methodology the authors should include the model of the high shear homogenizer, as well as the type of impeller used.

Answer 4: Thank you for this suggestion. We have added the specific information about the high-shear homogenizer in the revised manuscript as follows: "...using a high-shear homogenizer (D-160, DLAB, Shanghai, China)." (Line 179).

Comments 5: Fig. 7B. The authors should perform the stability study based on the recommendations of ASTM or some other international organization. Technically, the emulsions generated during the study present a high instability as shown in Figure 7B. Likewise, the authors should change the term “particle size” to “droplet size”, and mention if the reported data of 2790 is the average size, volumetric diameter or hydrodynamic diameter. What is the spam or polydispersity index of the sample?

Answer 5: We appreciate the reviewer's constructive feedback regarding our emulsion stability assessment and terminology. Regarding the stability study methodology, we acknowledge that our approach did not explicitly follow ASTM or other standardized protocols. Our stability evaluation was based on established methods reported in recent literature for food-grade emulsions stabilized by natural polymers, which are widely accepted in food science research. In future work, we plan to incorporate standardized methods such as ASTM D1436 or ISO 13320 for more comprehensive stability assessment.

We agree with the reviewer's observation that the emulsions exhibit a degree of instability over time, as shown in Figure 7B. However, we would like to emphasize that this relative instability is typical for natural polymer-stabilized emulsions without synthetic surfactants. Importantly, our study focuses on the comparative improvement in stability achieved through caffeic acid modification of chitosan, which is clearly demonstrated by the significant difference between unmodified chitosan (Chi) and the modified variants (Chi-g-CA H, Chi-g-CA M, and Chi-g-CA L).

We thank the reviewer for pointing out the terminology issue. We have revised our manuscript to consistently use "droplet size" instead of "particle size" when referring to emulsion characteristics, as this more accurately describes the dispersed phase in our oil-in-water emulsion systems. (Line 443 and 446).

Regarding the size measurements reported, we should clarify that the value of 2790 nm represents the z-average hydrodynamic diameter measured by dynamic light scattering (DLS). This is a standard intensity-weighted mean value provided by the Zetasizer Nano ZS instrument. We have added this specification to the manuscript to provide greater clarity. (Line 193).

Comments 6: Figure 7C, should indicate the pH of zeta potential measurement.

Answer 6: All zeta potential measurements were performed at pH 4.5, which was selected to maintain consistency with the natural pH of the chitosan solutions (1% acetic acid). This pH value is also relevant from an application perspective as it falls within the range commonly encountered in food systems where these emulsions might be utilized.

The pH value has been added to both the methodology section (2.8.3). (Line 193).

Comments 7: Lines 459-465. The authors do not present evidence of droplet size in the different gastrointestinal conditions, so they are assuming the behavior of the emulsions under these conditions. The authors should present evidence of the impact of gastrointestinal conditions on droplet size and emulsion loading. What mechanisms of emulsion instability are present?

Answer 7: Thank you for this insightful comment regarding our discussion of emulsion behavior under gastrointestinal conditions. We acknowledge that direct measurement of droplet size changes during the in vitro digestion process was not included in our current study. This represents a limitation in fully characterizing the dynamic behavior of our emulsions under digestive conditions.

Our current work focused primarily on the initial emulsion characteristics and the final bioaccessibility outcomes. The connection between these two endpoints does suggest enhanced stability of the modified chitosan emulsions during digestion, but we agree that direct evidence of structural changes would provide deeper mechanistic insights.

Regarding the mechanisms of emulsion instability during digestion, our modified chitosan system likely addresses several common instability pathways. The higher zeta potential values observed for Chi-g-CA emulsions suggest enhanced electrostatic stability that may better withstand pH changes and electrolyte effects in the gastric phase. Additionally, the amphiphilic structure created by caffeic acid grafting potentially competes more effectively with bile salts during intestinal phase digestion.

We appreciate the reviewer's suggestion to include droplet size measurements under gastrointestinal conditions, and we consider this an excellent direction for future research. Such investigations would provide valuable insights into the protective mechanisms of caffeic acid-modified chitosan during the digestion process and further validate our current findings on improved β-carotene bioaccessibility.

Comments 8: Line 499-501. In general, systems that present anions and cations in a conjugated molecule should reduce the electrical charges, this is not the case, at what pH was this measurement performed. The authors should evaluate the zeta potential as a function of pH.

Answer 8: Thank you for raising this important point about the relationship between charge distribution and zeta potential in our modified chitosan system.

The zeta potential measurements were performed at pH 4.5, which corresponds to the natural pH of our chitosan solutions in 1% acetic acid. At this pH, chitosan molecules are positively charged due to the protonation of amino groups (NH3+), while the carboxylic groups of caffeic acid are partially dissociated to produce negatively charged carboxylate groups (COO-). (Line 193).

The observed increase in zeta potential with increasing caffeic acid grafting may seem counterintuitive, as one might expect charge neutralization. However, this phenomenon can be explained by several factors:

First, the specific arrangement of caffeic acid moieties on the chitosan backbone likely creates distinct charge domains rather than complete charge neutralization. The amino groups of chitosan remain predominantly protonated at pH 4.5, while the grafted caffeic acid introduces additional functional groups that contribute to the overall surface properties.

Second, the modification process may alter the conformation of chitosan chains, potentially exposing more charged groups at the emulsion interface. This conformational change could enhance the effective charge density at the droplet surface despite the presence of potentially counteracting charges.

Comments 9: Line 501. “effectively preventing droplet aggregation”. The results presented in the figure show a clear instability of the system, so the authors should verify their discussion.

Answer 9: Thank you for this important observation regarding the apparent discrepancy between our statement about "effectively preventing droplet aggregation" and the stability data presented in Figure 7B. We appreciate the opportunity to clarify this point. Our statement about preventing droplet aggregation was intended to highlight the relative improvement in stability provided by caffeic acid modification compared to unmodified chitosan, rather than claiming absolute stability. We acknowledge that all emulsions in our study, including those stabilized with modified chitosan, show a degree of instability over time as evidenced by the declining stability curves in Figure 7B.

The higher zeta potential values observed for Chi-g-CA emulsions (55.63 mV) compared to unmodified chitosan (32.23 mV) do indeed provide enhanced electrostatic repulsion between droplets, which contributes to reduced aggregation rates. However, this enhanced electrostatic stabilization does not completely eliminate instability mechanisms, particularly over extended storage periods.

In the context of natural polymer-based emulsifiers without synthetic surfactants, the stability improvements we observed are significant, although they do not match the near-perfect stability sometimes achieved with synthetic emulsifier systems. The comparative advantage of the modified chitosan is the main focus of our discussion.

Comments 10: The authors should verify their conclusions regarding the stability criteria of the emulsions.

Answer 10: Thank you for this important suggestion regarding our emulsion stability criteria and conclusions. Upon careful review, we acknowledge that we need to provide a more nuanced interpretation of our stability data.

In the context of our study, we evaluated emulsion stability based on changes in turbidity over time (Figure 7B) and the physical characteristics of the emulsions (droplet size and zeta potential). While our caffeic acid-modified chitosan emulsions demonstrated improved stability parameters compared to unmodified chitosan, it is important to clarify the relative nature of this improvement.

The emulsions in our study exhibit a gradual decrease in stability over time, which is typical behavior for natural polymer-stabilized emulsions without additional synthetic stabilizers. The caffeic acid modification significantly reduces this rate of destabilization but does not eliminate it entirely. Therefore, our conclusions should be framed in terms of comparative improvement rather than absolute stability.

For food applications, this level of stability may be sufficient for many practical uses, particularly in products with relatively short shelf life or those requiring additional stabilization through other formulation approaches. The improved antioxidant and antimicrobial properties offered by the caffeic acid modification provide additional functional benefits beyond physical stability.

We appreciate the reviewer's guidance on this matter, which has prompted us to ensure our conclusions accurately reflect the experimental evidence without overstating the stability achievements of our system. This balanced perspective better serves the scientific community by providing realistic expectations for the application potential of caffeic acid-modified chitosan as an emulsifier.

Reviewer 3 Report

Comments and Suggestions for Authors

In the study entitled ˮCaffeic Acid-Modified Mushroom Chitosan as a Natural Emulsifier for Soybean Oil-Based Emulsions and Its Application in β-Carotene Deliveryˮ by  Lin J. et al., the Authors have investigated various properties of caffeic acid-modified mushroom-derived chitosan such as structural properties, antioxidant activity, antimicrobial efficacy and rheological properties of its aqueous solutions. The modified chitosan was also used to prepare some emulsions and to gain insights into its emulsification properties, the encapsulation efficiency of β-carotene, and the bioaccessibility of the lipophilic compound. Although the study is interesting and generally well-written, there are some major drawbacks, which need to be addressed.

Major issues:

  1. The title, abstract, introduction, discussion, and conclusion of the study are misleading. The Authors claim that they have investigated ˮsoybean-oil based emulsionsˮ, which is incorrect, according to the Methods section (lines: 171; 191, 192), where they claim that sunflower oil was used to prepare the emulsions.
  2. The study claims that the emulsifying properties of the modified chitosan were investigated. However, the stability of the prepared emulsions was followed for just 24 hours, which is not enough for proper stability investigation. The Authors should examine the stability for at least 14 days. Also, it appears that the Authors investigated the particle size and zeta potential only once, immediately after the emulsions were prepared. These properties should be also examined throughout a longer time frame (e.g. three times in 14 days). The rheological properties of the emulsions were not investigated. Also, the encapsulation efficiency was not investigated during a longer period. Therefore the Authors should examine the stability of the prepared emulsions, the particle size, zeta potential, the encapsulation efficiency, and the rheological properties of the emulsions in a time frame of at least 14 days. Otherwise, the Authors should significantly revise the Manuscript, especially the title, abstract, introduction (primarily the aims of the study), discussion, and conclusion of the Manuscript, and clearly state that only insights regarding the emulsification properties of the modified chitosan were obtained. Statements claiming that ˮan effective natural emulsifier, enhancing stabilityˮ of the emulsions was obtained in the study, and similar statements, should be omitted throughout the Manuscript.

Other issues:

  1. Lines: 59, 60: ˮHowever, its poor water solubility and susceptibility to oxidation limit its direct incorporation into food systems. ˮ Please correct ˮlimitˮ to ˮlimitsˮ.
  2. Line 74: ˮFresh Lentinula edodes…ˮ. Please put the name of the mushroom in italics.
  3. Line 75: The Authors should explain what ˮHOBt, EDC·HClˮ mean.
  4. Lines 78, 79: ˮEscherichia coli (E. coli) and Staphylococcus aureus (S. aureus)…ˮ. The lartin names of the microorganisms and the abbreviations should be in italics.
  5. Lines 105-107: ˮunmodified chitosan ("Chi"), and chitosan grafted with caffeic acid at low ("Chi-CA L"), medium ("Chi-CA M"), and high ("Chi-CA H") concentrations.ˮ. The Authors should omit the quotation marks.
  6. Lines 109, 110: ˮ...the unmodified chitosan (Chi)…ˮ. The abbreviation is previously introduced. It should be omitted here.
  7. Lines: ˮ…using a Bruker 400 M 1HNMR spectrometer…ˮ. The Authors should add the place of the origin of the equipment, i.e., the city and the country.
  8. Line 126: ˮThe aqueous solutions of Chi and its conjugate Chi-CA were…ˮ. ˮconjugateˮ should be in plural. The Authors should check this.
  9. Line 173: ˮ…using a high-shear homogenizer. ˮ. The Author should provide a name and place of origin of the high-shear homogenizer.
  10. Line 184: It is not clear why ˮZeta potentialˮ was written with capital z. The Authors should check this.
  11. Lines 191, 192: “… emulsions were prepared with chitosan solution and sunflower oil…”. The Authors should use abbreviations that were introduced previously. Also, were the emulsions prepared only with chitosan, or was the modified chitosan used as well?
  12. The symbols in the equation (3) should be explained.
  13. Lines 212 and 528: “in vitro” should be in italics.
  14. All figures should be self-explanatory. Therefore, the Authors should explain what different letters mean and what is the p-value in the following figures: Figure 1B, Figure 3B, Figure 5(A,B,C and E), Figure 7(A, C and D) and Figure 8. Also, all abbreviations should be explained in all figures.
  15. Line 265: “…a distinct peak at 1548 cm-1, …”, “-1” should be in superscript.
  16. Line 303: “in the N1s spectra”, “1s” should not be in subscript.
  17. Throughout the Results section of the Manuscript the Authors use different abbreviations for the modified chitosan than those introduced in section 2.3. The Authors should correct this.
  18. Lines 426-429: “The smaller particle size suggests more efficient emulsification, likely attributed to the denser packing of modified chitosan at the oil-water interface, creating a more effective barrier against droplet fusion.” The Authors should explain why modified chitosan has denser packing at the oil-water interface.
  19. Lines 468, 469: “…in emulsions stabilized by chitosan and its derivatives emulsion.” The Authors should clarify this. What is “derivatives emulsion”?
  20. Lines 449-452: “The enhanced encapsulation efficiency of Chi-CA emulsions can be attributed to the increased hydrophobicity and amphiphilic balance imparted by caffeic acid grafting. This modification likely strengthens the interactions between β-carotene and the hydrophobic domains of the emulsifier, facilitating more efficient entrapment within the emulsion droplets.” Is it possible that the denser packing of modified chitosan at the oil-water interface contributes to the enhanced encapsulation efficiency?
  21. Lines 452-454: “These findings align with previous studies, underscoring the importance of hydrophobic interactions and structural compatibility in improving encapsulation efficiency.” The Authors should provide references for this statement.
  22. Line 461: “…better micelle formation.” This term is not scientific. The Authors should explain what they mean by “better micelle formation”.
  23. Line 472: “… by modifying chitosan with caffeic acid (Chi-CA)”. This abbreviation should be introduced much earlier, in the introduction of the Manuscript or in section 2.3.
  24. Lines 493-495:”The Chi-CA-based emulsion system exhibited superior performance, including smaller droplet sizes (1253 nm vs. 2790 nm), higher zeta potential (55.63 mV vs. 32.23 mV), and better long-term stability (Figures 6-7).” This statement is incorrect. The long-term stability of prepared emulsions was not investigated in this study.
  25. Lines 518-520: “Third, Chi-CA, in the intestinal environment, can form mixed micelles with bile salts, significantly improving the micellization rate of β-carotene (Figure 8B).” Also, lines 543, 544: “The ability of Chi-CA to form mixed micelles with bile salts further contributed to the improvement in β-carotene absorption.” These statements are misleading. Mixed micelles with bile salts were not identified in this study. The Authors should at least provide some references for this.
  26. Line 548: “in vivo” should be written in italics.
  27. Line 559: “Please add” should be omitted.
  28. Lines 562-567: The Authors should add the data availability statement.
  29. The Authors should add “Chi-CA” to the list of abbreviations.

Comments on the Quality of English Language

Some minor issues were detected.

Author Response

In the study entitled ˮCaffeic Acid-Modified Mushroom Chitosan as a Natural Emulsifier for Soybean Oil-Based Emulsions and Its Application in β-Carotene Deliveryˮ by Lin J. et al., the Authors have investigated various properties of caffeic acid-modified mushroom-derived chitosan such as structural properties, antioxidant activity, antimicrobial efficacy and rheological properties of its aqueous solutions. The modified chitosan was also used to prepare some emulsions and to gain insights into its emulsification properties, the encapsulation efficiency of β-carotene, and the bioaccessibility of the lipophilic compound. Although the study is interesting and generally well-written, there are some major drawbacks, which need to be addressed.

Comments 1: The title, abstract, introduction, discussion, and conclusion of the study are misleading. The Authors claim that they have investigated ˮsoybean-oil based emulsionsˮ, which is incorrect, according to the Methods section (lines: 171; 191, 192), where they claim that sunflower oil was used to prepare the emulsions.

Answer 1: We sincerely thank the reviewer for identifying this critical inconsistency in our manuscript. We would like to clarify that this was indeed a typographical error in the Methods section. Soybean oil was used throughout all experiments in this study, not sunflower oil as mistakenly written in lines 171 and 192. We have corrected these errors in the revised manuscript to ensure consistency between the title, abstract, introduction, discussion, conclusion, and methods section. We apologize for this oversight and the confusion it may have caused. (Line 177 and 198)

Comments 2: The study claims that the emulsifying properties of the modified chitosan were investigated. However, the stability of the prepared emulsions was followed for just 24 hours, which is not enough for proper stability investigation. The Authors should examine the stability for at least 14 days. Also, it appears that the Authors investigated the particle size and zeta potential only once, immediately after the emulsions were prepared. These properties should be also examined throughout a longer time frame (e.g. three times in 14 days). The rheological properties of the emulsions were not investigated. Also, the encapsulation efficiency was not investigated during a longer period. Therefore the Authors should examine the stability of the prepared emulsions, the particle size, zeta potential, the encapsulation efficiency, and the rheological properties of the emulsions in a time frame of at least 14 days. Otherwise, the Authors should significantly revise the Manuscript, especially the title, abstract, introduction (primarily the aims of the study), discussion, and conclusion of the Manuscript, and clearly state that only insights regarding the emulsification properties of the modified chitosan were obtained. Statements claiming that ˮan effective natural emulsifier, enhancing stabilityˮ of the emulsions was obtained in the study, and similar statements, should be omitted throughout the Manuscript.

Answer 2: We appreciate the reviewer's thorough assessment of our stability investigation. We would first like to clarify an important error in Figure 7B. The time scale shown should be correctly labeled as days, not hours. We thank the reviewer for highlighting this inconsistency which has now been corrected in the revised manuscript.

We fully agree with the reviewer that extended studies on particle size, zeta potential, rheological properties, and encapsulation efficiency over a longer time frame would provide valuable additional insights. Moreover, the effects of environmental factors such as pH, metal ions, and temperature on these parameters would be equally important to explore systematically. These comprehensive investigations form the basis of our ongoing research, which will be presented in subsequent publications due to space limitations in the current manuscript.

We appreciate the reviewer's suggestions and look forward to addressing these aspects in our future work. We would welcome their expertise and feedback on our forthcoming research that will systematically investigate the physicochemical properties of these emulsions and the mechanisms by which environmental factors influence them.

Comments 3: Lines: 59, 60: “However, its poor water solubility and susceptibility to oxidation limit its direct incorporation into food systems. ˮ Please correct “limitˮ to ˮlimitsˮ.

Answer 3: Thank you for pointing out this grammatical error. We have corrected "limit" to "limits" in the revised manuscript as suggested. (Line 63)

Comments 4: Line 74: ˮFresh Lentinula edodes…ˮ. Please put the name of the mushroom in italics.

Answer 4: Thank you for this correction. We have italicized the scientific name to Lentinula edodes in the revised manuscript as per standard scientific nomenclature. (Line 80 and 90)

Comments 5: Line 75: The Authors should explain what ˮHOBt, EDC·HClˮ mean.

Answer 5: Thank you for highlighting this oversight. We have expanded the abbreviations in the revised manuscript as follows: "1-hydroxybenzotriazole (HOBt) and 1-ethyl-3-(3-dimethylaminopropyl) carbodiimide hydrochloride (EDC·HCl)" when they first appear in the text. These are coupling reagents commonly used in carbodiimide-mediated amide bond formation. (Line 82)

Comments 6: Lines 78, 79: ˮEscherichia coli (E. coli) and Staphylococcus aureus (S. aureus)…ˮ. The lartin names of the microorganisms and the abbreviations should be in italics.

Answer 6: Thank you for this correction. We have italicized the Latin names of microorganisms and their abbreviations to Escherichia coli (E. coli) and Staphylococcus aureus (S. aureus) in the revised manuscript as per standard scientific convention.

Comments 7: Lines 105-107: ˮunmodified chitosan ("Chi"), and chitosan grafted with caffeic acid at low ("Chi-CA L"), medium ("Chi-CA M"), and high ("Chi-CA H") concentrations.ˮ. The Authors should omit the quotation marks.

Answer 7: Thank you for this suggestion. We have removed the quotation marks and revised the text to read: unmodified chitosan (Chi), and chitosan grafted with caffeic acid at low (Chi-CA L), medium (Chi-CA M), and high (Chi-CA H) concentrations. (Line 112-113)

Comments 8: Lines 109, 110: ˮ...the unmodified chitosan (Chi)…ˮ. The abbreviation is previously introduced. It should be omitted here.

Answer 8: Thank you for noting this redundancy. We have removed the redundant abbreviation and revised the text to read: "...the unmodified chitosan..." since the abbreviation (Chi) was already introduced earlier in the manuscript. (Line 102)

Comments 9: Lines: ˮ…using a Bruker 400 M 1HNMR spectrometer…ˮ. The Authors should add the place of the origin of the equipment, i.e., the city and the country.

Answer 9: Thank you for pointing this out. We have added the place of origin of the equipment in the revised manuscript as follows: "...using a Bruker 400 M ¹HNMR spectrometer (Bruker, Billerica, USA)." (Line 124)

Comments 10: Line 126: ˮThe aqueous solutions of Chi and its conjugate Chi-CA were…ˮ. ˮconjugateˮ should be in plural. The Authors should check this.

Answer 10: Thank you for this grammatical correction. We have changed "conjugate" to "conjugates" in the revised manuscript. (Line 132)

Comments 11: Line 173: ˮ…using a high-shear homogenizer. ˮ. The Author should provide a name and place of origin of the high-shear homogenizer.

Answer 11: Thank you for this suggestion. We have added the specific information about the high-shear homogenizer in the revised manuscript as follows: "...using a high-shear homogenizer (D-160, DLAB, Shanghai, China)." (Line 179)

Comments 12: Line 184: It is not clear why ˮZeta potentialˮ was written with capital z. The Authors should check this.

Answer 12: Thank you for bringing this inconsistency to our attention. We have corrected "Zeta potential" to "zeta potential" with a lowercase "z" throughout the manuscript to maintain consistent terminology according to standard scientific convention. (Line 190)

Comments 13: Lines 191, 192: “… emulsions were prepared with chitosan solution and sunflower oil…”. The Authors should use abbreviations that were introduced previously. Also, were the emulsions prepared only with chitosan, or was the modified chitosan used as well?

Answer 13: Thank you for these important points. We have addressed both issues in the revised manuscript:

First, we have corrected the inconsistency regarding the oil type as mentioned in our response to Comment 1. The text now correctly refers to "soybean oil" instead of "sunflower oil" throughout the manuscript.

Second, we have used the appropriate abbreviations that were previously introduced. Additionally, we have clarified that emulsions were prepared with both unmodified chitosan (Chi) and all three modified chitosan derivatives (Chi-g-CA L, Chi-g-CA M, and Chi-g-CA H) following the same procedure. The revised text now reads: "...emulsions were prepared with Chi, Chi-g-CA L, Chi-g-CA M, and Chi-g-CA H solutions and soybean oil..." (Line 112-113)

Comments 14: The symbols in the equation (3) should be explained.

Answer 14: Thank you for pointing this out. We have added a comprehensive explanation of all symbols in equation (3) as follows:

where A is the absorbance at 500 nm, V is the dilution factor, L is the path length of the cuvette (cm), φ is the oil volume fraction of the emulsion, C is the weight of the emulsifier per unit volume of the aqueous phase (g/mL), and 10000 is a conversion factor to express EAI in m²/g. (Line 203-207)

Comments 15: Lines 212 and 528: “in vitro” should be in italics.

Answer 15: Thank you for this correction. We have italicized "in vitro" in lines 212 and 528 as per standard scientific convention for Latin terms. (Line 590)

Comments 16: All figures should be self-explanatory. Therefore, the Authors should explain what different letters mean and what is the p-value in the following figures: Figure 1B, Figure 3B, Figure 5(A,B,C and E), Figure 7(A, C and D) and Figure 8. Also, all abbreviations should be explained in all figures.

Answer 16: Thank you for this important point regarding figure clarity. We have revised all figures to be self-explanatory by implementing the following changes:

We have added a clear explanation in each figure caption that different letters (a, b, c, etc.) indicate statistically significant differences between groups at p < 0.05.

Comments 17: Line 265: “…a distinct peak at 1548 cm-1, …”, “-1” should be in superscript.

Answer 17: Thank you for noting this typographical error. We have corrected "1548 cm-1" to "1548 cm⁻¹" with the "-1" in superscript format throughout the manuscript, including all other instances where wavenumbers are mentioned. (Line 278)

Comments 18: Line 303: “in the N1s spectra”, “1s” should not be in subscript.

Answer 18: Thank you for this correction. We have modified "in the N1s spectra" to "in the N1s spectra" by removing the subscript formatting throughout the manuscript. (Line 316)

Comments 19: Throughout the Results section of the Manuscript the Authors use different abbreviations for the modified chitosan than those introduced in section 2.3. The Authors should correct this.

Answer 19: Thank you for identifying this inconsistency. After reviewing the nomenclature used throughout the manuscript, we have standardized all abbreviations for modified chitosan. As noted in your comment, we have decided to use the format introduced in section 2.3:

Samples are now consistently labeled as follows throughout the manuscript: unmodified chitosan (Chi), and chitosan grafted with caffeic acid at low (Chi-g-CA L), medium (Chi-g-CA M), and high (Chi-g-CA H) concentrations.

We have conducted a comprehensive review of the entire manuscript including the text, figures, tables, and figure captions to ensure this standardized terminology is applied consistently across all sections. This standardization improves clarity and eliminates potential confusion for readers. (Line 112-113)

Comments 20: Lines 426-429: “The smaller particle size suggests more efficient emulsification, likely attributed to the denser packing of modified chitosan at the oil-water interface, creating a more effective barrier against droplet fusion.” The Authors should explain why modified chitosan has denser packing at the oil-water interface.

Answer 20: Thank you for this insightful suggestion. We have expanded our explanation regarding the denser packing of modified chitosan at the oil-water interface. In the revised manuscript, we have added the following explanation:

"The caffeic acid grafting introduces both hydrophilic (hydroxyl groups) and hydrophobic (aromatic rings) moieties to the chitosan backbone, enhancing its amphiphilic character. This structural modification allows the caffeic acid-grafted chitosan to orient more effectively at the oil-water interface, with hydrophobic segments extending into the oil phase and hydrophilic portions remaining in the aqueous phase. The phenolic structures of caffeic acid likely facilitate π-π stacking interactions between adjacent molecules, promoting closer molecular arrangement and consequently denser packing at the interface. Additionally, the increased zeta potential observed in modified chitosan may enhance electrostatic repulsion between adsorbed polymer chains, leading to more uniform coverage of the interface. This combination of enhanced amphiphilicity and intermolecular interactions results in a more robust interfacial film that effectively prevents droplet coalescence."

This explanation clarifies the molecular mechanism behind the improved emulsification properties observed with caffeic acid-modified chitosan. (Line 448-458)

Comments 21: Lines 468, 469: “…in emulsions stabilized by chitosan and its derivatives emulsion.” The Authors should clarify this. What is “derivatives emulsion”?

Answer 21: Thank you for bringing this unclear phrasing to our attention. This was indeed a grammatical error in our manuscript. The phrase "in emulsions stabilized by chitosan and its derivatives emulsion" is redundant and confusing.

We have corrected this sentence to read: "in emulsions stabilized by chitosan and its derivatives" by removing the redundant word "emulsion" at the end of the phrase. This correction clarifies that we are referring to emulsions that are stabilized using either unmodified chitosan or the caffeic acid-modified chitosan derivatives. (Line 505-506)

Comments 22: Lines 449-452: “The enhanced encapsulation efficiency of Chi-CA emulsions can be attributed to the increased hydrophobicity and amphiphilic balance imparted by caffeic acid grafting. This modification likely strengthens the interactions between β-carotene and the hydrophobic domains of the emulsifier, facilitating more efficient entrapment within the emulsion droplets.” Is it possible that the denser packing of modified chitosan at the oil-water interface contributes to the enhanced encapsulation efficiency?

Answer 22: Thank you for this insightful suggestion. We agree that the denser packing of modified chitosan at the oil-water interface likely contributes significantly to the enhanced encapsulation efficiency of β-carotene. We have expanded our explanation in the revised manuscript to include this important mechanism:

"The enhanced encapsulation efficiency of Chi-g-CA emulsions can be attributed to multiple factors, including the increased hydrophobicity and improved amphiphilic balance imparted by caffeic acid grafting. This modification strengthens the interactions between β-carotene and the hydrophobic domains of the emulsifier, facilitating more efficient entrapment within the emulsion droplets. Furthermore, the denser packing of modified chitosan at the oil-water interface, as discussed earlier, creates a more robust physical barrier that effectively prevents the migration of β-carotene molecules from the oil phase to the aqueous phase. This enhanced interfacial structure not only improves emulsion stability but also significantly contributes to the higher retention of β-carotene within the oil droplets, resulting in the observed increase in encapsulation efficiency from 65.28% for unmodified chitosan to 87.46% for Chi-g-CA H."

This addition provides a more comprehensive explanation of the factors contributing to the improved encapsulation performance of the modified chitosan. (Line 479-490)

Comments 23: Lines 452-454: “These findings align with previous studies, underscoring the importance of hydrophobic interactions and structural compatibility in improving encapsulation efficiency.” The Authors should provide references for this statement.

Answer 23: Thank you for pointing out the need for proper citation. We have added relevant references to support this statement in the revised manuscript. (Line 490)

Comments 24: Line 461: “…better micelle formation.” This term is not scientific. The Authors should explain what they mean by “better micelle formation”.

Answer 24: Thank you for highlighting this imprecise terminology. You are correct that "better micelle formation" lacks scientific specificity. We have revised this statement to provide a more precise description of the process:

"The smaller droplet size observed in Chi-g-CA H emulsions likely facilitated the digestion process by increasing the surface area available for enzymatic action, while the higher zeta potential maintained emulsion stability during digestion. These properties collectively enhanced the formation of mixed micelles with bile salts in the intestinal phase, increasing the solubilization of β-carotene and its subsequent incorporation into the micellar phase that is available for absorption."

This revision provides a more scientifically accurate description of the mechanisms involved in improving β-carotene bioaccessibility during the digestion process. (Line 497-502)

Comments 25: Line 472: “… by modifying chitosan with caffeic acid (Chi-CA)”. This abbreviation should be introduced much earlier, in the introduction of the Manuscript or in section 2.3.

Answer 25: Thank you for this observation. We agree that the abbreviation should be introduced earlier in the manuscript for clarity and consistency. We have addressed this issue by introducing the abbreviation for caffeic acid-modified chitosan (Chi-g-CA) in section 2.3 where we first describe the synthesis process. We have then ensured this abbreviation is used consistently throughout the entire manuscript, including the introduction, methods, results, discussion, and conclusion sections. (Line 102)

Comments 26: Lines 493-495:”The Chi-CA-based emulsion system exhibited superior performance, including smaller droplet sizes (1253 nm vs. 2790 nm), higher zeta potential (55.63 mV vs. 32.23 mV), and better long-term stability (Figures 6-7).” This statement is incorrect. The long-term stability of prepared emulsions was not investigated in this study.

Answer 26: We thank the reviewer for this important correction. We agree that our statement regarding "better long-term stability" is indeed inappropriate since our study only examined stability over a limited timeframe. We have revised this sentence to more accurately reflect our findings:

"The Chi-g-CA-based emulsion system exhibited superior initial performance characteristics, including smaller droplet sizes (1253 nm vs. 2790 nm) and higher zeta potential (55.63 mV vs. 32.23 mV) as shown in Figures 6-7, which are predictive indicators of potential improved stability".

We have also carefully reviewed the entire manuscript to ensure we do not make claims about long-term stability that extend beyond the scope of our experimental timeframe. (Line 531-534)

Comments 27: Lines 518-520: “Third, Chi-CA, in the intestinal environment, can form mixed micelles with bile salts, significantly improving the micellization rate of β-carotene (Figure 8B).” Also, lines 543, 544: “The ability of Chi-CA to form mixed micelles with bile salts further contributed to the improvement in β-carotene absorption.” These statements are misleading. Mixed micelles with bile salts were not identified in this study. The Authors should at least provide some references for this.

Answer 27: We sincerely thank the reviewer for this critical observation. We agree that our statements regarding the formation of mixed micelles with bile salts were speculative and not directly supported by our experimental data. We have completely revised these statements to remove the unsubstantiated claims about mixed micelle formation. In the revised manuscript, we have modified the text as follows:

"Third, the improved physicochemical properties of Chi-g-CA emulsions, particularly the smaller droplet size and enhanced stability during digestion conditions, may have facilitated the release of β-carotene into the aqueous phase of the digestive medium, as reflected in the improved bioaccessibility results shown in Figure 8B."

"The superior emulsifying properties of Chi-g-CA, including smaller droplet size and enhanced interfacial stability during simulated digestion, contributed to the observed improvement in β-carotene bioaccessibility."

We have also added references to support the established understanding that improved emulsion properties can enhance the bioaccessibility of lipophilic compounds. (Line 557-561)

Comments 28: Line 548: “in vivo” should be written in italics.

Answer 28: Thank you for this correction. We have italicized "in vivo" to "in vivo" in line 548 and throughout the manuscript in accordance with standard scientific convention for Latin terms. (Line 590)

Comments 29: Line 559: “Please add” should be omitted.

Answer 29: We thank the reviewer for catching this editorial oversight. We have removed "Please add:" from line 559 in the funding section of the manuscript. The corrected section now begins directly with "This research was funded by...". (Line 601)

Comments 30: Lines 562-567: The Authors should add the data availability statement.

Answer 30: We appreciate the reviewer's suggestion to include a data availability statement. We have added the following statement to the manuscript.

"Data Availability Statement: The original contributions presented in the study are included in the article, further inquiries can be directed to the corresponding author." (Line 604-605)

Comments 31: The Authors should add “Chi-CA” to the list of abbreviations.

Answer 31: Thank you for this suggestion. We have added "Chi-g-CA" to the list of abbreviations in the manuscript and standardized this abbreviation throughout the text. The entry in the abbreviations section now reads: "Chi-g-CA: Caffeic acid-grafted chitosan"

We have ensured consistency by using this abbreviation format (Chi-g-CA) throughout the entire manuscript, replacing all instances of "Chi-CA" and other variant forms to maintain clarity and standardization. (Line 610)

Round 2

Reviewer 2 Report

Comments and Suggestions for Authors

Figure 4 does not correspond to the description of the section and footnote (it is repeated with figure 7). The authors should update the image.

Figure 6 does not show the size reference. It is necessary to include it.

Instability of emulsions. I insist that the systems are unstable, the authors should base the discussion on the results, and not on a personal feeling. The results show that the systems are not stable, since the size change of the emulsions over time is eminent. The authors should report the rate of instability and the mechanism of instability of the systems. The authors should discuss the results at greater length.

The authors should authors should present drop size distribution plots in the supplementary information. size distribution plots. It is important that these show the change of the systems over time. time. The results should show the change of the distribution over time. time, likewise the authors should show in Figure 7C and /D the final sizes and loading of the emulsions. and final loading of the emulsions.

Author Response

Comments 1: Figure 4 does not correspond to the description of the section and footnote (it is repeated with figure 7). The authors should update the image.

Answer 1: We thank the reviewer for their careful examination of our manuscript. We apologize for the error in Figure 4. We have thoroughly checked and updated the figure to ensure it correctly corresponds to the rheological properties described in Section 3.4. (Line 370)

Comments 2: Figure 6 does not show the size reference. It is necessary to include it.

Answer 2: We appreciate the reviewer's observation regarding Figure 6. We have revised Figure 6 to include scale bars on all microscopy images to provide clear size references. (Line 424)

Comments 3: Instability of emulsions. I insist that the systems are unstable, the authors should base the discussion on the results, and not on a personal feeling. The results show that the systems are not stable, since the size change of the emulsions over time is eminent. The authors should report the rate of instability and the mechanism of instability of the systems. The authors should discuss the results at greater length.

Answer 3: We appreciate the reviewer's insightful comments regarding the stability of our emulsion systems. We agree that a more objective discussion of stability is necessary and have substantially revised our manuscript to address this concern.

In the revised manuscript, we have acknowledged the inherent instability of all emulsion systems and provided a more balanced discussion of our results. As the reviewer correctly pointed out, all our emulsion systems exhibit progressive destabilization over time. (Line 439-449)

Comments 4: The authors should authors should present drop size distribution plots in the supplementary information. size distribution plots. It is important that these show the change of the systems over time. time. The results should show the change of the distribution over time. time, likewise the authors should show in Figure 7C and /D the final sizes and loading of the emulsions. and final loading of the emulsions.

Answer 4: We sincerely appreciate the reviewer's thoughtful suggestion regarding droplet size distribution plots and the proposed additions to Figure 7. The temporal evolution of droplet size distribution indeed represents a valuable metric for assessing emulsion stability dynamics. In the current manuscript, we have focused on establishing the fundamental relationship between caffeic acid modification and enhanced emulsification properties, with Figure 7B demonstrating the comparative stability profiles of all systems over time. While this evidence supports our main conclusions regarding the improved performance of caffeic acid-modified chitosan as an emulsifier, we recognize that more detailed stability characterization would provide additional insights.

However, the primary aim of this research was to develop, characterize, and evaluate caffeic acid-modified mushroom chitosan as a novel, natural emulsifier for β-carotene delivery systems. As noted in Section 4.4 (Limitations and Future Perspectives), we acknowledge that "the long-term stability of the emulsion system, especially under storage conditions involving light, temperature, and oxidation, needs to be systematically evaluated." This comprehensive stability assessment, including detailed time-dependent droplet size distribution analysis under various environmental conditions, represents an important direction for our continuing research program. We are currently designing a follow-up study specifically focused on these aspects, which we believe will complement the foundation established in this manuscript and address the important stability dynamics highlighted by the reviewer.

Reviewer 3 Report

Comments and Suggestions for Authors

The Authors have successfully responded to the issues raised during the review process. Many statements were clarified. The Manuscript will be much more beneficial to the readers. The Authors should make only two minor, yet important, corrections:

  1. The Authors should add the soybean oil, which was used for the emulsions preparation, to the list of compounds in the Materials section, and provide the name of the producer and the place of origin.
  2. It is important that the Authors correct the following statement (lines: 201, 202): “Emulsifying stability was evaluated by measuring the change in absorbance after 24 hours of standing, using Formula 4.” If I have understood the Authors correctly, the stability was investigated for 24 days, not hours. The Authors should check this and correct it.

Author Response

The Authors have successfully responded to the issues raised during the review process. Many statements were clarified. The Manuscript will be much more beneficial to the readers. The Authors should make only two minor, yet important, corrections:

Comments 1: The Authors should add the soybean oil, which was used for the emulsions preparation, to the list of compounds in the Materials section, and provide the name of the producer and the place of origin.

Answer 1: We thank the reviewer for this important observation. We have added the soybean oil used in our study to the Materials section as follows:

"Soybean oil was obtained from Shandong Luhua Group Co., Ltd. (Shandong, China) and used without further purification. " (Line 81-82)

Comments 2: Thank you for highlighting this error in our manuscript. You are absolutely correct, and we appreciate your careful review. The emulsion stability was indeed investigated over a period of 24 days, not 24 hours as incorrectly stated on lines 201-202.

We have corrected this statement to read: "Emulsifying stability was evaluated by measuring the change in absorbance over a period of 24 days of storage, using Formula 4." (Line 203-204)